# Lactational delivery of Triclosan promotes non-alcoholic fatty liver disease in newborn mice

André A. Weber[1], Xiaojing Yang[1], Elvira Mennillo[1], Jeffrey Ding[2], Jeramie D. Watrous[2], Mohit Jain[2], Shujuan Chen[1], Michael Karin[3] & Robert H. Tukey ®[1] ✉

Here we show that Triclosan (TCS), a high-volume antimicrobial additive that has been detected in human breastmilk, can be efficiently transferred by lactation to newborn mice, causing significant fatty liver (FL) during the suckling period. These findings are relevant since pediatric non-alcoholic fatty liver disease (NAFLD) is escalating in the United States, with a limited mechanistic understanding. Lactational delivery stimulated hepatosteatosis, triglyceride accumulation, endoplasmic reticulum (ER) stress, signs of inflammation, and liver fibrosis. De novo lipogenesis (DNL) induced by lactational TCS exposure is shown to be mediated in a PERK-eIF2α-ATF4-PPARα cascade. The administration of obeticholic acid (OCA), a potent FXR agonist, as well as activation of intestinal mucosal-regenerative gp130 signaling, led to reduced liver ATF4 expression, PPARα signaling, and DNL when neonates were exposed to TCS. It is yet to be investigated but mother to child transmission of TCS or similar toxicants may underlie the recent increases in pediatric NAFLD.

Non-alcoholic FLD (NAFLD) has emerged as the most common liver disorder worldwide, paralleling the obesity and diabetes epidemics[1]. The prevalence of NAFLD in adults in Western countries is estimated between 20 and 30% and approximately 10 and 20% of these cases have a more severe manifestation named non-alcoholic steatohepatitis (NASH)[2]. Although NASH usually appears in adults, there is an alarming increase in NAFLD in children, termed pediatric NAFLD[3,4], which is a prerequisite towards NASH development. Epidemiological studies have shown a prevalence of pediatric NAFLD in up to 10% of the pediatric population[5]. NAFLD in children displays the same morphological lesions observed in adults, however, the distribution of these lesions is frequently different. For example, in adult's, steatosis starts in the perivenular zone, while in children steatosis starts in the periportal zone[6]. The progression of simple steatosis to NASH depends on multiple parallel hits, including endoplasmic reticulum (ER) stress, inflammation, defective lipid export, enhanced de novo lipogenesis (DNL) and deterioration of the intestinal barrier[7–10].

Physiological changes and different pathological conditions can activate an elaborate signaling pathway called integrated stress response (ISR). Extrinsic (hypoxia, amino acid deprivation, and viral infection) and intrinsic (ER) stresses can activate ISR. ER stress, which plays an important role in fatty liver disease (FLD), activates the activating transcription factor 4 (ATF4) through protein kinase R like kinase (PERK)[11]. Peroxisome proliferator-activated receptor α (PPARα) is also a key regulator of hepatic fatty acid oxidation (FAO) and lipid metabolism[12]. However, the involvement of PPARα in the progression of FLD is controversial[13,14]. Moreover, PPARα has a strong link to ER stress signaling. Atf4-null mice challenged with a high fructose diet have exhibited diminished lipogenesis and PPARα target gene expression[15]. PPARα also has a strong link with ATF6, another ER sensor. Overexpression of ATF6 in NAFLD livers leads to activation of the ATF6-PPARα axis, promoting an increase in hepatic FAO genes including fibroblast growth factor 21 (Fgf21), a liver-secreted cytokine that regulates hepatic metabolic processes, including fat oxidation,

[1]Laboratory of Environmental Toxicology, Department of Pharmacology, University of California, San Diego, La Jolla, CA 92093, USA. [2]Departments of Medicine and Pharmacology, University of California, San Diego, La Jolla, CA 92093, USA. [3]Laboratory of Gene Regulation and Signal Transduction, Department of Pharmacology, University of California, San Diego, La Jolla, CA 92093, USA. ✉e-mail: rtukey@health.ucsd.edu

gluconeogenesis, and metabolic gene expression[16]. FGF21 has been shown to ameliorate NAFLD[17].

Environmental toxicants in the absence of caloric overload can induce FLD or toxicant-associated FLD (TAFLD)[18,19]. Triclosan (TCS) is a high-volume chemical used as an antimicrobial additive in many human consumer products[20,21]. We have demonstrated that long-term TCS exposure in adult mice increased liver-to-body weight (BW) without affecting BW[22]. This resulted in enhanced liver proliferation along with induction of genes linked to fibrogenesis, elevated collagen accumulation, and liver oxidative stress, supporting the notion that TCS may lead to a condition similar to NASH, referred to as toxicant associated steatohepatitis (TASH). We have recently confirmed that TCS induces FLD while blunting HFD-induced expression of FGF21[23].

TCS is a ubiquitous environmental toxicant[21] and has been identified in human breastmilk samples with concentrations up to 2100 μg/ kg of lipids[24,25]. Based upon the volume of breastmilk that an infant consumes per day it has been estimated that a breastfeeding infant can consume 1000–2000 ng of TCS daily[24,26]. In addition, while breastmilk has an important influence on infant survival and health reducing disease risk and promoting aspects of postnatal development[27], environmental toxicants can still be transferred from the nurturing mother to newborns through lactation[28]. The constant presence of TCS in breastmilk in different studies and the high incidence of pediatric NAFLD led us to hypothesize that the transfer of TCS from lactating mothers to newborns can lead to NAFLD in children. To examine the effect of TCS presented to neonatal mice through lactation, we exposed pregnant females to TCS in their diet and evaluated the delivery of TCS through breastmilk.

Here we show the delivery of TCS through lactation leads to the precocious development of fatty liver suggesting that exposure of newborns to TCS and other toxicants through lactation may be a contributing factor to early onset NAFLD and NASH.

## Results

### TCS stimulates hepatic ER stress and DNL in mice

Mating C57/BL6 mice were placed on a normal chow diet containing 0.012% TCS that was continued after birth with neonates exposed through this route for 21 days. TCS in milk and serum of breastfed neonates was measured by LC-MS/MS analysis. The concentration of TCS in breastmilk isolated from the stomach of neonates was $81 \pm 19.5$ μg/kg at day 14, while the serum concentration at 21 days of exposure was $52.8 \pm 8.5$ μg/kg (Fig. 1a). These concentrations were like those reported in milk and serum from human samples[25,29]. Breastfeeding with TCS has no effect on body weight and liver weight (Supplementary Fig. S1a).

Neonatal exposure to TCS through breastmilk resulted in upregulation of ER stress in liver, reflected by induction of the genes glucose-regulated protein 78 (*Grp78*), ER degradation enhancing alpha-mannosidase like protein 1 (*Edem1*), spliced X-box binding protein-1 (*Xbp1s*) and C/EBP homologous protein (*Chop*), with increases in activated phosphorylated eukaryotic translation initiation factor 2α (p-eIF2α) and activating transcription factor 4 (ATF4) (Fig. 1b).

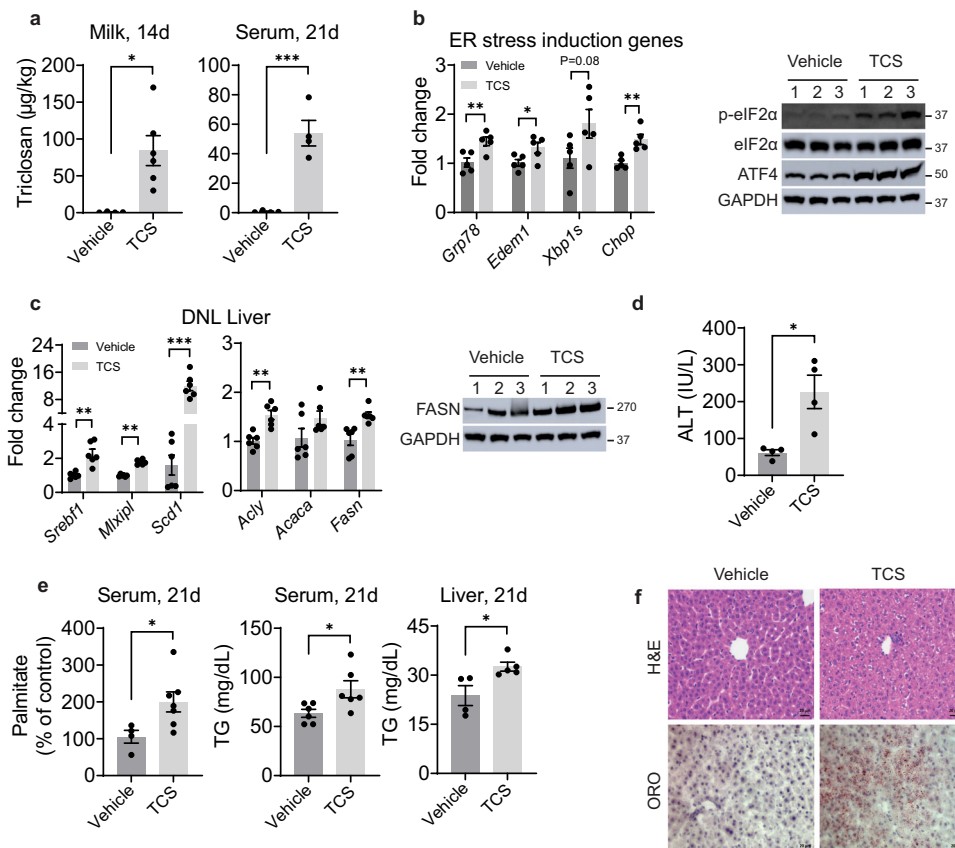

**Fig. 1 | Triclosan exposure induces ER stress, stimulates DNL and causes TG accumulation.** WT mice breastfed with TCS or vehicle for 14 or 21 days. **a** TCS was measured in milk from stomach contents of 14-day old mice (Vehicle $n = 4$ and TCS $n = 6$ per group) and serum at 21 days ($n = 4$ per group). **b** Expression of hepatic ER stress genes ($n = 5$ per group) and immunoblot (IB) analysis of total and phosphorylated eIF2α and ATF4 in liver lysate from 21-day old mice ($n = 3$ mice per group). **c** Expression of hepatic lipogenic genes at 21 days ($n = 6$ mice per group) and IB analysis of FASN in livers ($n = 3$ mice per group). **d** Measurement of ALT in serum at 21 days old ($n = 4$ per group) **e** Measurement of serum palmitate (%), vehicle, $n = 4$ and TCS, $n = 7$, serum, $n = 6$ per group and liver triglycerides (TG), vehicle, $n = 4$ and TCS, $n = 5$ at 21 days. **f** Histology liver sections were stained with H&E and ORO ($n = 4$ per group), Scale bars=20μm. (**a**–**e**) show mean ± S.E., determined by two-tailed Student's test; *$P < 0.05$, **$P < 0.01$, ***$P < 0.001$.

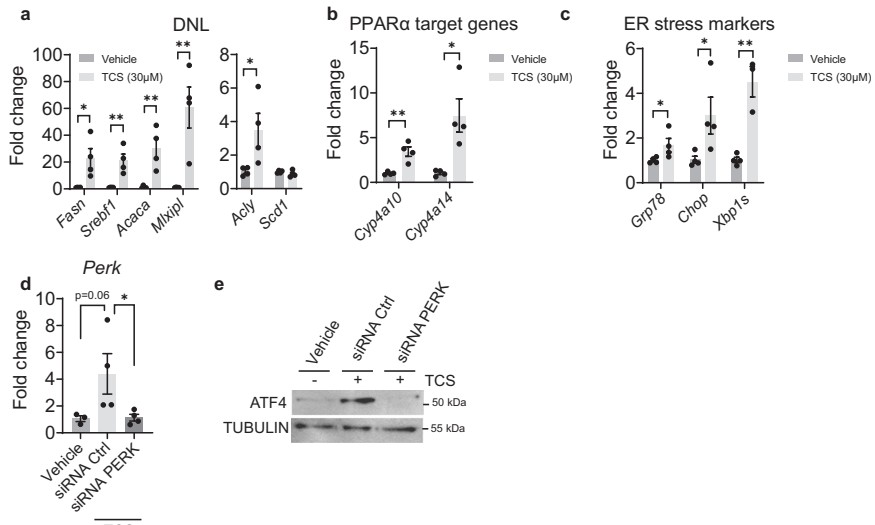

**Fig. 2 | TCS stimulates DNL, ER stress and PPARα in mouse primary hepatocytes and ATF4 induction is PERK dependent.** Mouse primary hepatocytes were isolated from one month old mice and were placed in medium with DMSO or in the presence of 30 μM of TCS. After 72 h of exposure the hepatocytes were collected for Q-RT-PCR analysis. **a** TCS stimulates genes associated with DNL, *n* = 4 samples per group. **b** Downstream PPARα target genes, *n* = 4 samples per group. **c** Induction of genes associated with ER stress, *n* = 4 samples per group. **d** Hepatocytes were isolated, and cells were transfected with mouse PERK siRNA and control siRNA. Twenty-four hours later, TCS at 30 μM was added in the medium and 48-h later cells were harvested. Q-RT-PCR analysis of *Perk* after treatment with siRNA (vehicle, *n* = 3; siRNA Ctrl, *n* = 4 and siRNA PERK, *n* = 4). **e** Representative IB analysis were performed for ATF4. (**a**–**d**) show mean ± S.E., determined by two-tailed Student's test; *\*P* < 0.05, \*\**P* < 0.01, \*\*\**P* < 0.001.

Examination of the hepatic gene profile indicated that neonatal mice breastfed with TCS exhibited increased expression of genes that participate in DNL, including sterol regulatory element-binding protein 1 (*Srebf1*), carbohydrate-responsive element-binding protein (ChREBP, encoded by *Mlxipl*), ATP citrate lyase (*Acly*), acetyl-CoA carboxylase (*Acaca*), fatty acid synthase (*Fasn*) and stearoyl-CoA desaturase-1 (*Scd1*) (Fig. 1c). However, there were no statistical differences between vehicle and TCS in expression of fatty acid β-oxidation genes, including peroxisomal acyl-coenzyme A oxidase 1 (*Acox1*) and carnitine palmitoyl transferase I (*Cpt1a*) (Supplementary Fig. 1b). Through immunoblot analysis (IB), FASN, which is a key enzyme in DNL[30], was also increased in livers of TCS breastfed mice (Fig. 1c). Furthermore, the increase in serum alanine aminotransferase (ALT) in TCS exposed mice is an indication that the liver has been damaged (Fig. 1d). By LC-MS/MS, palmitate in serum of neonates was two-fold higher than in control-fed neonatal mice (Fig. 1e). Congruently, liver and serum triglycerides (TG) were elevated in neonatal mice exposed to TCS (Fig. 1e). Oil Red O (ORO) staining showed accumulation of lipid droplets in the cytoplasm of hepatocytes from TCS-treated neonatal mice (Fig. 1f). However, Ki-67 staining and TUNEL assay revealed no effects of TCS on proliferation and cell death, respectively (Supplementary Fig. S1c, d).

## TCS stimulates ER stress, DNL and PPARα in primary hepatocytes

To investigate the effects of TCS in vitro, we treated primary hepatocytes with TCS at 30 μM for 72 h. After treatment, TCS was shown to upregulate genes driving DNL including *Fasn*, *Acaca*, *Mlxipl*, *Acly* and *Srebf1* (Fig. 2a), in addition to PPARα target genes (*Cyp4a10* and *Cyp4a14*) (Fig. 2b). Significant induction was also observed with genes associated with ER stress (*Grp78*, *Xbp1s*, and *Chop*) (Fig. 2c). These findings suggest that a strong correlation exists between TCS exposure and ER stress. The induction of genes linked to ER stress in vivo and in vitro led us to speculate that ER stress may be linked to PERK activation followed by ATF4 activation. To examine this possibility, we first treated primary hepatocytes with PERK-specific siRNA. Q-RT-PCR analysis confirmed downregulation of *Perk* gene expression (Fig. 2d).

In the absence of PERK-specific siRNA, ATF4 was induced by TCS, as shown by IB analysis. However, in the presence of PERK-specific siRNA, TCS treatment did not induce ATF4 (Fig. 2e). These findings indicate that PERK activation and induction of ATF4 underlie the processes leading to ER stress and induction of DNL.

## TCS induces NAFLD and HCC-related genes

Utilizing RNA-seq analysis to investigate those pathways in neonate liver impacted by lactational TCS delivery, results showed that TCS exposure was associated with alterations in more than 410 upregulated genes and 532 downregulated genes, with an adjusted *P* value less than 0.05 (Supplementary Fig. S2). Gene ontology profiles revealed that TCS robustly induced genes linked to lipogenesis, fatty acid uptake, cholesterol biosynthesis and acylglycerol metabolism. Another set of genes increased by TCS treatment are the major urinary protein (*Mup*) genes, including, *Mup17*, *Mup19*, *Mup15*, *Mup9*, and *Mup3* (Fig. 3a). TCS exposure also increased genes associated with retinol metabolism, including cytochrome P450 26A1 (*Cyp26a1*), dehydrogenase/reductase SDR family member 4 (*Dhrs4*) and 9 (*Dhrs9*), aldehyde dehydrogenase family 1, subfamily A7 (*Aldh1a7*) and patatin-like phospholipase domain-containing protein 3 (*Pnpla3*) (Fig. 3a), which has an important role in NAFLD disease progression[31].

Previous results in our laboratory have demonstrated that long-term TCS exposure can promote hepatocellular carcinoma (HCC)[22]. RNA-seq analysis demonstrated that mini-chromosome maintenance (*Mcm*) genes are upregulated in TCS-treated mice, which have been reported in several types of cancer, including HCC[32]. Examination of the MCM genes indicated that TCS exhibited an increase in MCM −2, −4, −5, −6, −7, −8, and −10. Several other genes related to HFD-induced HCC[33] that were statistically upregulated by TCS exposure included Golgi membrane protein 1 (*Golm1*), integrin alpha-6 (*Itga6*) and ephrin A1 (*Efna1*) (Fig. 3a). However, many of genes linked to the onset of HCC-related genes were not statistically regulated by TCS treatment (Supplementary Fig. S3).

Progression from NAFLD to NASH is accompanied by fibrosis, inflammation, and oxidative stress[7,8,23,31]. In the present study, fibrogenic, oxidative stress, and inflammation marker genes were

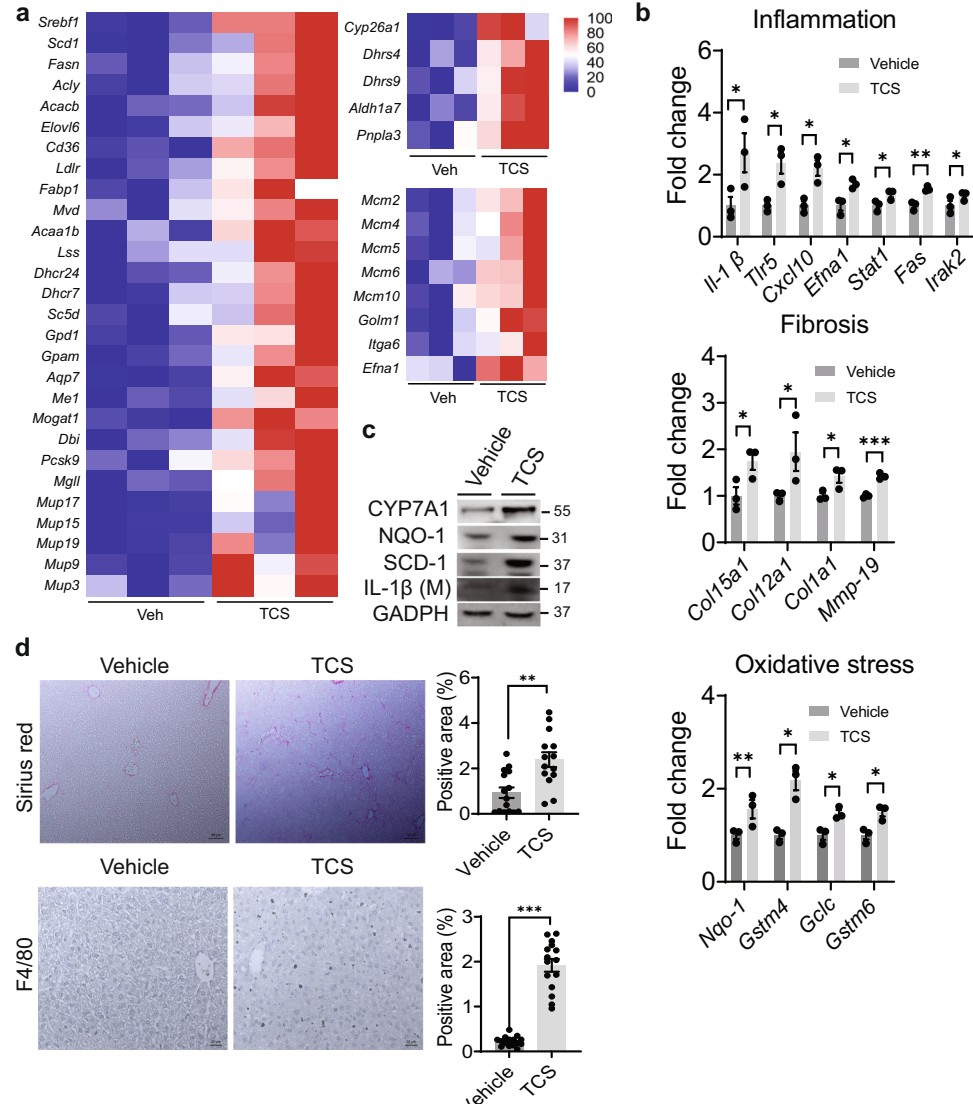

**Fig. 3 | TCS induces genes related to NASH and HCC in liver.** WT mice breastfed with TCS or vehicle for 21 days. **a** Heatmaps showing differential expression of genes related to lipogenesis, retinol metabolism and MCM and HCC in 21-day old mice (*n* = 3 per group). **b** Expression of genes associated with inflammation, fibrotic and oxidative stress (*n* = 3 per group). **c** Representative IB analysis of NQO1, SCD1 and IL-1β at 21 days (*n* = 1 per group). **d** Histological slides were stained with Sirius Red and F4/80 antibody in liver sections from TCS breastfed (21-day old) or vehicle breastfed WT mice, *n* = 15 images per group from 4 mice in each group. Sirius Red scale bars=50 μm and F4/80 scale bars = 20 μm (**b**) and (**d**) show mean ± S.E., determined by two-tailed Student's test; *$P < 0.05$, **$P < 0.01$, ***$P < 0.001$. All genes used are significant and *p* adjusted value below the cut-off level of 0.05.

upregulated in TCS-exposed neonatal mice (Fig. 3b). The increase in CYP7A1, NQO-1, SCD1, and IL-1β were confirmed by IB analysis (Fig. 3c). Sirius red staining and F4/80 immunohistochemistry showed increases in fibrosis and immune infiltration in TCS-treated neonatal mice (Fig. 3d). A semi-quantitative assessment score according to immune infiltration, fibrosis, steatosis, and cell death in the liver was evaluated in each treatment group. According to the range of each lesion the score can go from 0 to 12[34]. The analysis indicates a score of 3 in TCS breastfed mice and 0 for neonatal mice breastfed in the absence of TCS exposure (Supplementary Table S1 and Fig. S4). There was no evidence of hepatocyte ballooning or necrosis.

**TCS blocks hepatic glucocorticoid response**

TCS exposure abrogated the response to glucocorticoids in the liver. Glucocorticoids activate the glucocorticoid receptor (GR) and regulate important metabolic pathways in the liver, including gluconeogenesis[35], adenosine monophosphate (AMP)–activated protein kinase (AMPK)[36] and the urea cycle[37]. RNA-seq analysis demonstrated that genes related to gluconeogenesis and the urea cycle were significantly downregulated in livers of TCS breastfed mice. Furthermore, gene and protein analysis showed a decrease in the expression of forkhead box O1 (*Foxo1*), an important transcription factor involved in gluconeogenesis[38] (Supplementary Fig. S5a–d).

We have previously confirmed that TCS disrupted FGF21 expression[23]. GR can control AMPK target genes via FGF21[36]. Q-RT-PCR and ELISA analysis revealed downregulation of FGF21, in both liver and serum of TCS exposed neonatal mice (Fig. 4a). Surprisingly, IB analysis showed decreased levels of phosphorylated AMPK in TCS exposed mice. From RNA-seq analysis, AMPK target genes including peroxisome proliferator-activated receptor γ coactivator 1α (*Ppargc1a*), mitochondrial fission factor (*Mff*), unc-51 like autophagy activating kinase 1 (*Ulk1*) and glycogen Synthase 2 (*Gys2*) were also downregulated (Fig. 4b). Other hepatokines including fetuin B (*Fetub*),

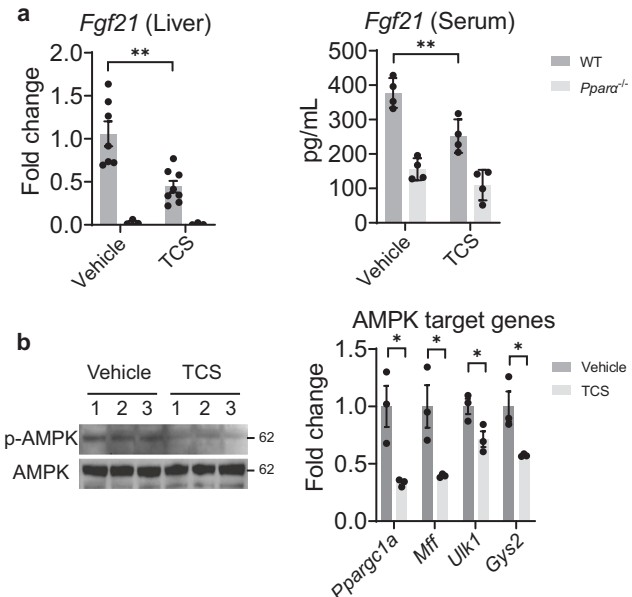

**Fig. 4 | TCS blocks *Fgf21* expression and AMPK signaling.** WT mice breastfed with TCS or vehicle for 21 days. **a** Hepatic expression of *Fgf21* was measured by Q-RT-PCR, vehicle WT, $n = 7$; vehicle *Pparα*$^{-/-}$, $n = 3$; TCS WT, $n = 8$; TCS *Pparα*$^{-/-}$, $n = 3$. and FGF21 serum levels were evaluated by ELISA in 21-day old mice, $n = 4$ per group. **b** IB analysis of total and phosphorylated AMPK and expression of AMPK target genes from TCS and WT mice in liver of 21-day old neonatal mice ($n = 3$ per group). (**a**) and (**b**) show mean ± S.E., determined by two-tailed Student's test; *$P < 0.05$, **$P < 0.01$. AMPK target genes used are significant and $p$ adjusted value below the cut-off level of 0.05.

retinol binding protein 4 (*Rbp4*), fetuin A (*Ahsg*), selenoprotein P (*Selenop*) and growth/differentiation factor 15 (*Gdf15*) (Supplementary Fig. S5e) were downregulated following TCS exposure.

## ATF4 controls DNL in TCS-exposed mice

Q-RT-PCR and IB analysis confirmed the absence of ATF4 in the liver of *Atf4*$^{ΔHep}$ mice (Supplementary Fig. S6a). When we examined expression of genes linked to DNL in *Atf4*$^{F/F}$ and *Atf4*$^{ΔHep}$ mice exposed to lactational TCS, Q-RT-PCR analysis showed that induction of *Srebf1*, *Mlxipl*, *Acaca* and *Fasn* genes, all associated with the development of DNL, were abrogated in *Atf4*$^{ΔHep}$ neonatal mice exposed to TCS (Fig. 5a). Consistent with these results, TCS exposed *Atf4*$^{ΔHep}$ mice had less TG in their livers compared to *Atf4*$^{F/F}$ mice (Fig. 5b). *Atf4*$^{ΔHep}$ mice also expressed less FASN (Fig. 5c) and showed reduced TCS-induced lipid accumulation (Fig. 5d).

ATF4 interacts with many transcription factors[11]. To analyze a possible link between ATF4 and PPARα, we examined PPARα target genes in *Atf4*$^{F/F}$ and *Atf4*$^{ΔHep}$ mice treated with TCS. Q-RT-PCR analysis showed decreased expression of *Cyp4a10* and *Cyp4a14* in *Atf4*$^{ΔHep}$ mice relative to *Atf4*$^{F/F}$ mice (Supplementary Fig. S6b). Furthermore, blockage of PERK in primary hepatocytes with PERK siRNA followed by TCS treatment demonstrated that when the PERK-ATF4 axis is blocked, PPARα target genes (*Cyp4a10*, *Cyp4a14*, and *Fabp1*) and several genes associated with DNL (*Srebf1*, *Fasn*, *Acaca* and *Mlxipl*) are down-regulated (Fig. 5e). These findings indicate that PPARα acts downstream of ATF4 to control DNL.

## PPARα controls DNL in TCS-treated mice

Liver tissues from NASH patients exhibit induction of PPARα and its downstream target genes[33]. RNA-seq analysis confirmed that PPARα signaling was activated in livers of TCS-exposed neonatal mice (Supplementary Fig. S7). Examining PPARα target gene expression by Q-RT-PCR we found robust upregulation of downstream genes, including

*Cyp4a14*, *Ehhadh,* and *Cyp4a10* (Fig. 6a). Induction of hepatic *Cyp4a14* by PPARα has been linked to FLD[15].

In addition, PPARα activation by TCS upregulated genes linked to DNL, such as *Mlxipl*, *Acly*, *Acaca*, *Fasn*, *Srebf1*, and *Scd1*, whose induction by TCS was blocked in *Pparα*$^{-/-}$ mice (Fig. 6b). IB analysis confirmed reduced expression of FASN and SCD1 proteins in livers of *Pparα*$^{-/-}$ mice exposed to TCS (Fig. 6c). ELISA showed reduced accumulation of TG in TCS exposed *Pparα*$^{-/-}$ livers (Fig. 6d). Histological analysis of *Pparα*$^{-/-}$ mice exposed to TCS was comparable with that of *Atf4*$^{ΔHep}$ mice in which hepatosteatosis was blocked (Fig. 6e). Combined, these results indicate that PPARα is required for induction of hepatosteatosis in TCS exposed neonatal mice.

## Obeticholic acid (OCA) blocks TCS-induced DNL

Lactational delivery of TCS to neonatal mice resulted in reduced expression of hepatic FXR target genes (Supplementary Fig. S8a). FXR agonists, including OCA, were developed as potential therapeutics for NAFLD and NASH[39–41]. Neonatal mice nursing on normal milk or TCS tainted breastmilk were treated daily from postnatal day 16 to 20 with oral OCA (100 mg/kg) and tissues collected on postnatal day 21. OCA treatment resulted in robust induction of *Shp*, an FXR target gene in liver and small intestine (Fig. 7a and Supplementary Fig. S8b). FXR activation was reported to enhance expressions of intestinal barrier genes, such as claudins and tight-junction proteins (TJPs)[42]. However, our results revealed few changes in intestinal tissue of TCS-exposed neonates treated with OCA, other than decreased *Tjp1*, *Cldn1,* and *Cldn19* mRNAs (Supplementary Fig. S8c).

OCA administration had a significant impact on liver. Q-RT-PCR analysis demonstrated that OCA reduced the expression of genes driving DNL including *Mlxipl*, *Acaca*, *Fasn*, and *Scd1* (Fig. 7b). OCA also repressed PPARα target genes, including *Cyp4a10*, *Cyp4a14,* and *Fabp1*, but only when mice were exposed to TCS (Fig. 7c). The treatment with OCA increases *Fgf21* gene expression even when mice are exposed to TCS (Fig. 7d). ELISA showed reduced hepatic accumulation of TG in mice treated with OCA along with TCS (Fig. 7e). OCA administration along with TCS decreases *Grp78* and *Xbp1s* (Fig. 7f), while IB analysis showed reduced ATF4 and FASN proteins in OCA treated neonatal mice exposed to TCS (Fig. 7g). Importantly, ORO staining revealed a decrease in lipid accumulation in mice treated with OCA along with TCS (Fig. 7h).

OCA administration also had an impact on the GR and AMPK response. Q-RT-PCR analysis demonstrated that OCA increased the expression of metallothionein 2 (*Mt2*) and Sulfotransferase Family 1E Member 1 (*Sult1e1*), important GR target genes, only in the absence of TCS. We can conclude that OCA does not stimulate GR targets in the presence of TCS. However, *Ppargc1a*, a key AMPK target gene, plays an important role in FAO and helps to limit lipid accumulation in the liver[43]. While the *Ppargc1a* gene is downregulated by TCS (Fig. 4), OCA overrides this repression by showing considerable induction in the presence of TCS (Supplementary Fig. S8d), an event that may contribute towards protecting the liver from TCS-induced FLD.

## Gp130 signaling blocks TCS-induced DNL

An increase in intestinal barrier permeability is one of the factors underlying the progression of NAFLD[10]. TCS treatment damaged the intestinal barrier, as demonstrated by accelerated apoptosis with increases in cytochrome C and induction of caspase-9 and caspase-3 (Fig. 8a). Apoptosis was mirrored by downregulation of claudin mRNAs (*Cldn1* and *Cldn19*) (Fig. 8b). Moreover, the upregulation of several antimicrobial proteins (AMPs), regenerating islet-derived protein 3 beta and gamma (*Reg3b* and *Reg3g*) and resistin-like beta (*Retnlb*), suggested that TCS is causing stress or microbial translocation in the intestinal mucosa (Supplementary Fig. S9a). Furthermore, TCS administration dramatically increased phosphorylated total β-catenin and its targets LGR5 and SGK1 (Supplementary Fig. S9b). β-catenin is

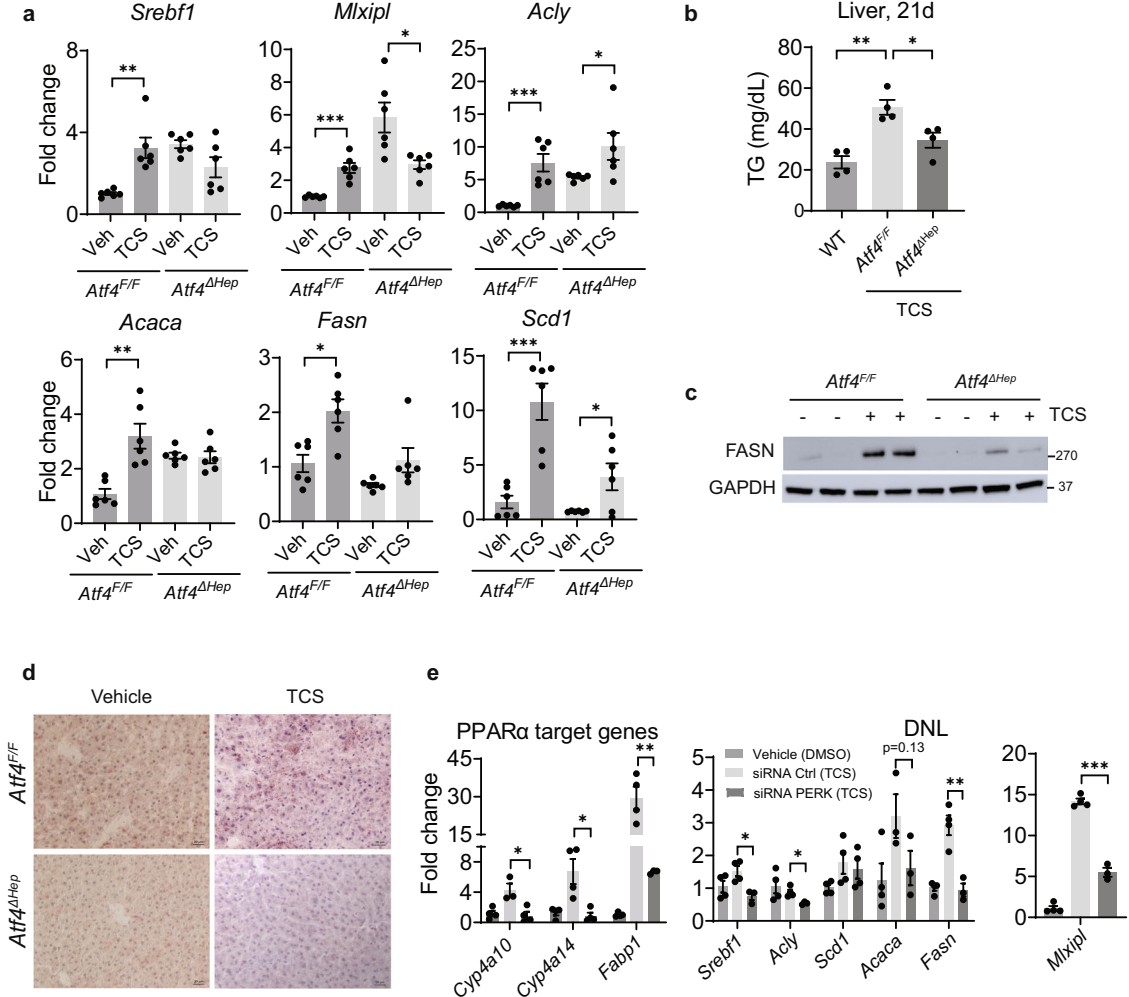

**Fig. 5 | ATF4 controls DNL in liver of TCS-treated mice.** $Atf4^{F/F}$ and $Atf4^{\Delta Hep}$ mice breastfed with vehicle or through lactational delivery of TCS for 21 days. **a** Expression of hepatic lipogenic genes, $n = 6$ mice per group. **b** Measurement of liver triglycerides (TG), $n = 4$ mice per group. **c** IB analysis of FASN in liver lysate from 21-day old mice ($n = 3$ mice per group). **d** Frozen liver sections were stained with ORO ($n = 3$ per group). Scale bars, 20μm. **e** Expression of target genes regulated by PPARα and those that participate in DNL following PERK siRNA and TCS treatment, vehicle, $n = 4$, siRNA Ctrl, $n = 4$ and siRNA PERK, $n = 3$. (**a**–**e**) show mean ± S.E., determined by two-tailed Student's test; *$P < 0.05$, **$P < 0.01$.

known to promote tumorigenesis via the *Wnt* pathway[44]. However, histological analysis did not present any tumor formation (Supplementary Fig. S9c). With the induction of intestinal AMPs, TCS administration is causing significant stress without altering the majority of the claudins and TJPs mRNAs (Supplementary Fig. S9d, e).

Intestinal damage activates yes-associated protein-1 (YAP) and tafazzin (TAZ) to stimulate barrier repair[45]. Indeed, TCS exposure upregulated YAP target genes, including connective tissue growth factor (*Ctgf*) in the small intestine and colon and cyclin e1 (*Ccne1*) in the small intestine (Fig. 8c). Due to tight-junction loss and apoptosis caused by TCS, we examined the actions of lactational TCS exposure on neonatal IEC-specific $gp130^{Act}$ mice which show enhanced intestinal barrier repair and upregulation of TJP and claudins[10]. Q-RT-PCR analysis showed that $gp130^{Act}$ mice were resistant to TCS-induced *Fasn*, *Scd1*, *Acaca*, *Acly* and *Mlxipl* mRNAs, all of which play important roles in DNL (Fig. 8d). In addition, PPARα target mRNAs, including *Cyp4a10* and *Fabp1* were reduced in livers of TCS exposed $gp130^{Act}$ mice (Fig.8e). Hepatic TG levels in $gp130^{Act}$ mice treated were also reduced when compared to WT treated mice (Fig. 8f). In addition, mRNA levels of *Grp78* were lower in livers of $gp130^{Act}$ mice when compared to WT treated with TCS (Fig. 8g). IB analysis showed lower protein levels of ATF4, FASN and SCD-1 in $gp130^{Act}$ mice compared with WT mice both treated with TCS (Fig. 8h). Importantly, ORO staining analysis showed

no accumulation of lipid droplets in vehicle and TCS exposed $gp130^{Act}$ mice when compared to WT mice exposed to TCS (Fig. 8i). Clearly, $gp130^{Act}$ mice are resistant to the early actions of lactational TCS delivery on the development of FL, clearly demonstrating that TCS impacts intestinal barrier integrity.

## Discussion

TCS, a ubiquitous environmental toxicant that has been detected in urine, blood, and breast milk in different regions of the world suggests that the general population is exposed to TCS[25,42,46,47]. With considerable evidence that TCS alters biological responses[21], several key studies in mice have confirmed that long-term exposure has detrimental effects on both the intestinal tract and liver. Chronic exposure to TCS has been shown to increase colonic inflammation and colitis-associated colon tumorgenesis[48], which has recently been linked to reactivation of TCS from its glucuronide metabolite by specific intestinal microbial β-glucuronidase enzymes[49]. Our previous studies have shown TCS can function as a liver tumor promoter stimulating liver tumorigenesis, due in part to the induction of oxidative stress and fibrosis[21,22]. In both a normal chow diet and high fat diet (HFD), TCS increased lipid droplet accumulation in liver[23]. As an adaptive response to HFD and nutrient sensing, the liver-secreted cytokine FGF21 is significantly induced. However, an HFD + TCS greatly blunts expression

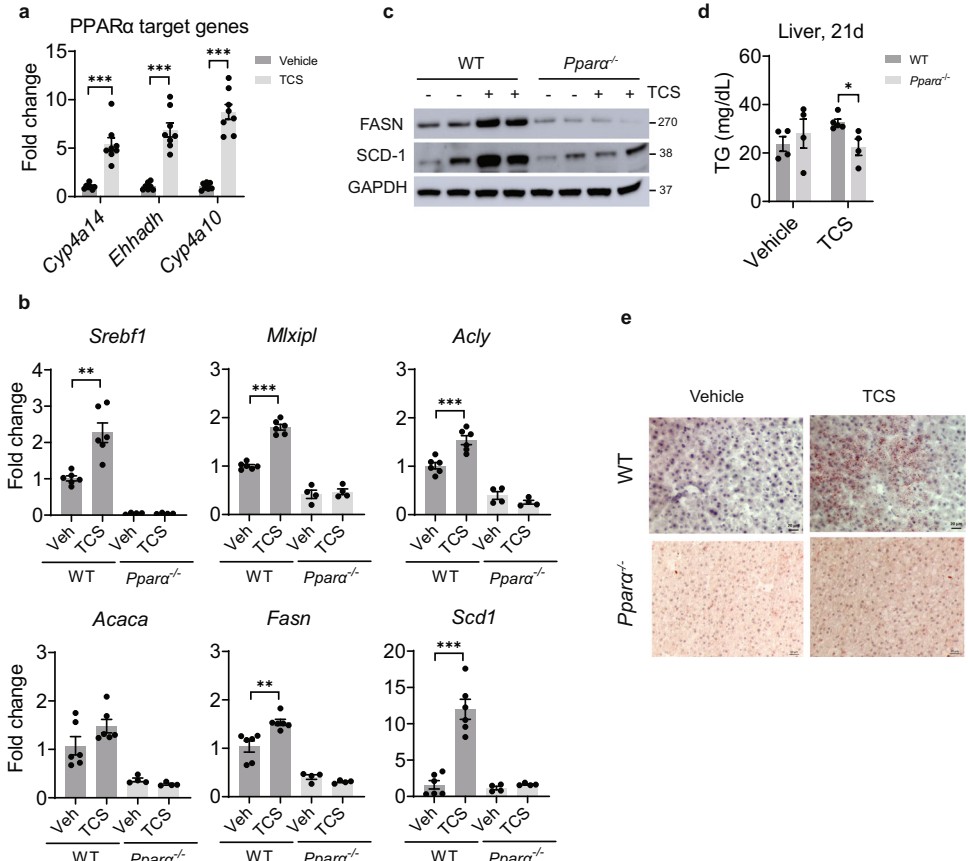

**Fig. 6 | PPARα controls DNL in liver of mice exposed to lactational TCS.** WT and *Pparα*[-/-] neonatal mice breastfed with vehicle or TCS for 21 days. **a** Expression of PPARα target genes at 21 days, *n* = 6 mice per group. **b** Expression of genes linked to hepatic lipogenesis, WT, *n* = 6 per group and *Pparα*[-/-], *n* = 4 per group. **c** Hepatic IB analysis of FASN and SCD1 (*n* = 2 mice per group). **d** Measurement of liver triglycerides (TG) at 21 days, *n* = 4 mice per group. **e** Frozen liver sections were stained with ORO (*n* = 3 per group). Scale bar, 20μm. (**a**, **b** and **d**) show mean ± S.E., determined by two-tailed Student's test; *P < 0.05, **P < 0.01, ***P < 0.001.

of FGF21[23], leading us to hypothesize that FGF21 plays an important role in suppressing FL. When we induced a type-1 diabetic animal model by treating newborn mice with streptozotocin to damage insulin-secreting beta cells in the pancreas[50] followed by a HFD, adult mice robustly developed NASH that was significantly accelerated in the presence of TCS[23]. TCS-treated mice exhibited elevated levels of oxidative stress, hepatic fibrosis, and inflammatory responses. With current epidemiological studies convincingly showing the accumulation of TCS in human breast milk, if it is transferred to nursing newborns the dietary conditions would simulate an early dietary-based HFD contaminated with TCS. Thus, we developed an animal model that allowed us to examine the impact of lactational TCS delivery to neonates and its potential impact on FLD.

Neonatal mice receiving TCS through lactation exhibit early onset hepatic ER stress and steatosis, two important events that stimulate progression from NAFLD to NASH[7]. Several mechanisms can lead to the induction of DNL in liver ER stress, including caspase-2 activation[51], inflammation[10], interleukin-17A[52] and the PERK-eIF2α-ATF4 axis[53]. Our findings indicate that ATF4 and PPARα are critical mediators of this process. Ablation of ATF4 or blockage with PERK siRNA prevents DNL-related gene induction in neonatal mice receiving TCS through lactation. Interestingly, deletion of PPARα also prevented lipid accumulation in the liver. We had previously shown that PPARα is activated by TCS through an indirect mechanism, as TCS had no effect on PPARα in vitro[22]. Similar outcomes regarding gene expression, TG accumulation and protein expression in both ATF4 and PPARα deficient mice suggest that activation of PPARα depends on ER stress through the PERK-eIF2α-ATF4 pathway.

AMPK has recently emerged as an important factor in NAFLD, due to its ability to control multiple metabolic pathways, including hepatic lipid metabolism[54]. Like HFD[54,55], TCS also downregulates hepatic AMPK signaling. AMPK reactivation blocks lipogenesis through ACC inhibition, decreasing intracellular malonyl-CoA levels, a precursor of triglycerides[54]. TCS also downregulates *Ppargc1a* and *Mff*, important factors involved with mitochondrial biogenesis and fission, respectively. NAFLD progression has been linked to decreases in *Ppargc1a* levels, a transcription factor involved in fatty acid oxidation[56].

Due to the role PPARα has on FAO, this nuclear receptor has been extensively studied as a potential target to treat NAFLD[57]. However, we have shown that PPARα ablation is protective in TCS-induced FLD. Furthermore, *Fgf21*, which is a PPARα target gene and plays an important role in fat oxidation[57] is downregulated by TCS treatment, unlike what is seen in HFD-fed mice[23]. Polychlorinated biphenyls (PCBs) have been shown to also reduce hepatic *Fgf21* mRNA[19]. Since TCS and other environmental toxicants can reduce FGF21 expression, NAFLD caused by environmental toxicants can be more aggressive than the development of NAFLD caused by HFD. However, the exact mechanism by which TCS downregulates FGF21 remains obscure.

OCA has been tested as a treatment option for several liver diseases including NAFLD and type-2 diabetes[41]. However, there is limited mechanistic understanding of how OCA acts. Our findings demonstrated that the oral administration of OCA dramatically reduces ATF4, PPARα signaling, and several genes associated with DNL in mice exposed to TCS. In addition, activation of gp130 signaling, using the *gp130*[Act] mouse model, prevented hepatic DNL and ER stress in TCS-

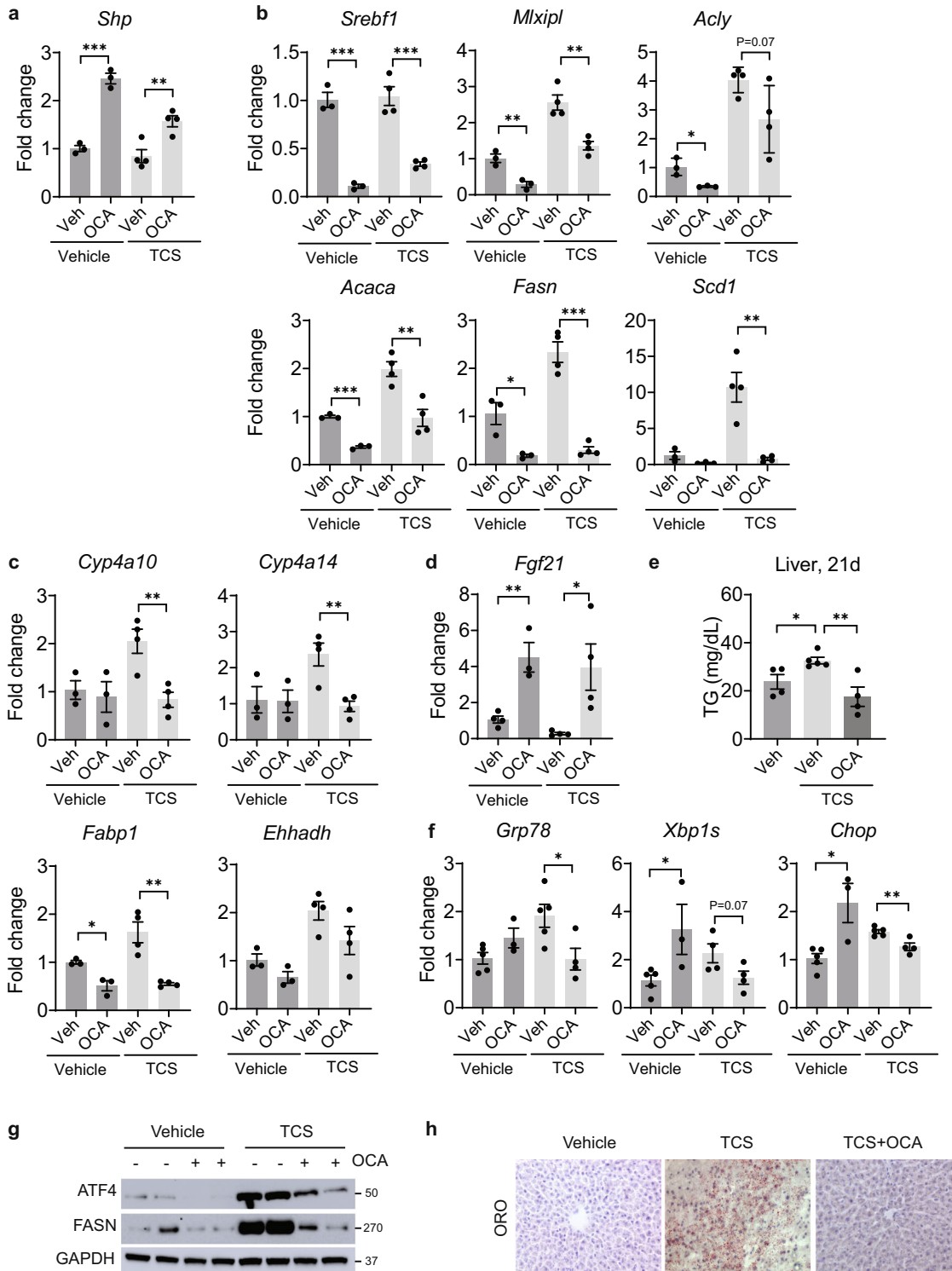

**Fig. 7 | Orally administration of OCA suppresses DNL genes, PPARα activity and ER stress in liver of TCS breastfed mice.** Neonatal mice receiving a normal diet or TCS through lactation for 21 days. During the course of this exposure, 16-day old mice were treated with OCA for 4 days and tissues collected on day 21. **a** Expression of hepatic *Shp* (vehicle, $n = 3$ per group and TCS, $n = 4$ per group). **b** Expression of genes associated with lipogenesis (vehicle, $n = 3$ per group and TCS, $n = 4$ per group). **c** Hepatic expression of PPARα target genes (vehicle, $n = 3$ per group and

TCS, $n = 4$ per group). **d** Hepatic expression of *Fgf21* (vehicle veh, TCS veh and TCS OCA, $n = 4$ per group and vehicle OCA, $n = 3$). **e** Measurement of liver TG ($n = 4$ per group). **f** Genes linked to ER stress (vehicle veh and TCS veh, $n = 5$ per group, TCS OCA, $n = 4$ and vehicle OCA, $n = 3$). **g** Hepatic IB analysis of ATF4 and FASN ($n = 2$ per group). **h** Frozen liver sections were stained with ORO ($n = 3$ per group). Scale bar, 20μm. (**a**–**f**) show mean ± S.E., determined by two-tailed Student's test; *$P < 0.05$, **$P < 0.01$, ***$P < 0.001$.

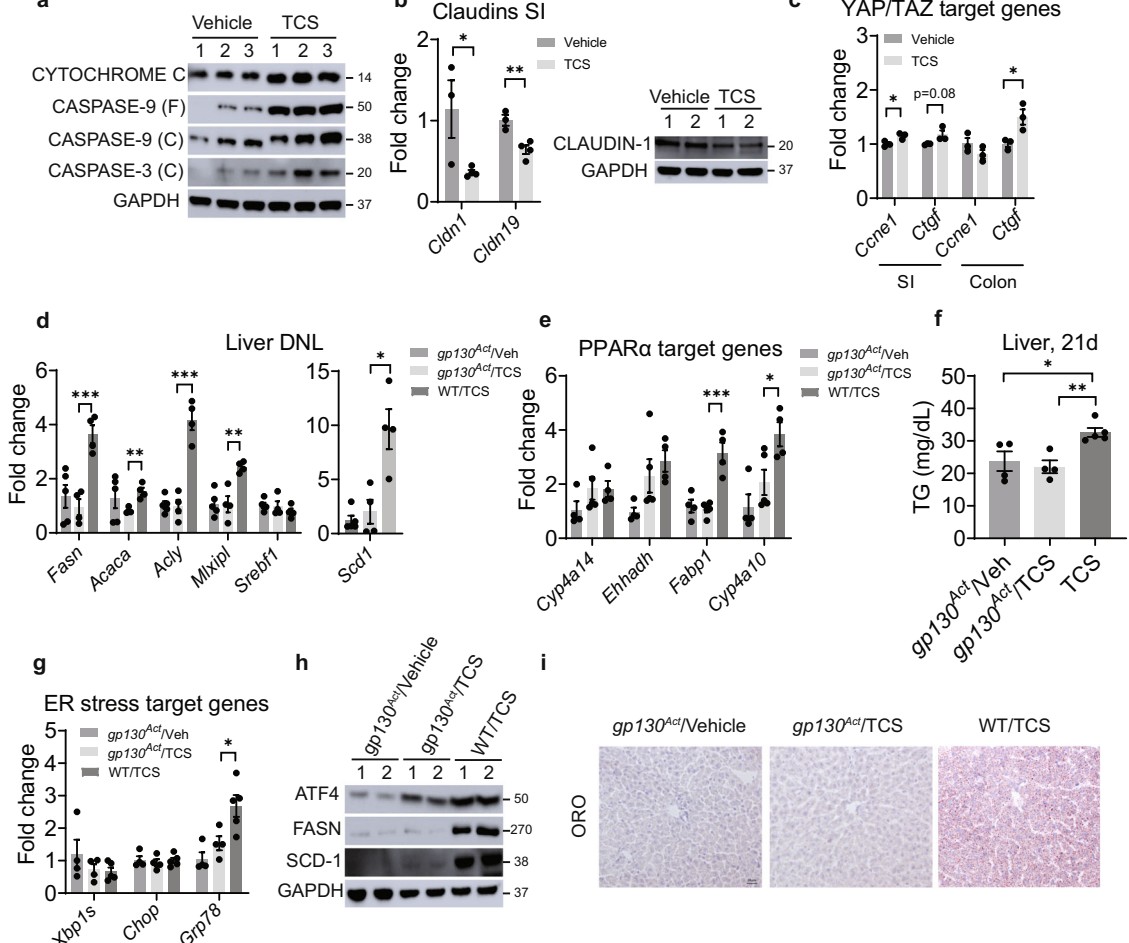

**Fig. 8 | Activation of gp130 suppresses liver DNL, PPARα activity and ER stress in liver of TCS breastfed mice.** *gp130*$^{Act}$ mice received normal milk or TCS through lactation for 21 days. **a** IB analysis of cytochrome C, caspase 9, cleaved caspase-3 (*n* = 3 per group). **b** Expression of claudins and IB of claudin-1 in intestines (vehicle, *n* = 3 per group and TCS, *n* = 4 per group). **c** Expression of YAP/TAZ target genes in the intestines (*n* = 3 per group). **d** Expression of hepatic lipogenic genes (*gp130*$^{Act}$/veh and WT, *n* = 4 mice per group and *gp130*$^{Act}$/TCS, *n* = 5). **e** Expression of

hepatic PPARα target genes (*gp130*$^{Act}$/veh and WT/TCS, *n* = 4 mice per group and *gp130*$^{Act}$/TCS, *n* = 5). **f** Measurement of liver TG (*gp130*$^{Act}$/veh and *gp130*$^{Act}$/TCS, *n* = 4 mice per group and WT/TCS, *n* = 5). **g** Hepatic genes linked to ER stress (*gp130*$^{Act}$/veh and *gp130*$^{Act}$/TCS, *n* = 4 mice per group and WT/TCS, *n* = 5). **h** Representative IB analysis of ATF4, FASN and SCD1 (*n* = 2 per group). **i** Frozen liver sections were stained with ORO (*n* = 3 per group). Scale bar, 20μm. (**b**–**g**) and (**f**) show mean ± S.E., determined by two-tailed Student's test; *P < 0.05, **P < 0.01, ***P < 0.001.

exposed mice. It is understood that gp130 reduces endotoxemia and decreases liver inflammation, thus decreasing fatty liver progression[10].

In summary, the transmission of TCS through lactation leads to lipogenesis, ER stress, PPARα activation and immune inflammation in neonatal livers. These events are important for early onset FLD. Mechanistically, both ATF4 and PPARα have an important role in DNL and NAFLD progression. The similarity in function between these two transcription factors suggests they may communicate physically. Further studies are necessary to elucidate this pathway. Administration of OCA and activation of mucosal-regenerative gp130 signaling ameliorates TCS-induced FLD, decreasing ER stress, PPARα signaling and DNL. As a ubiquitous environmental toxicant that has been detected in tissues from individuals of all ages[21], TCS may initiate TASH in children. In addition, with the increase in pediatric NAFLD and NASH[3,4], we can suggest that TCS exposure through lactation heightens the sensitivity toward FLD as children are continually exposed to high caloric and energy rich foods.

## Methods

### Mice and treatment options
All mice were housed in a specific pathogen–free facility with a 12-h light, 12-h dark cycle and given free access to food and water. Breeding

pairs of C57/B6 and *Pparα*-null (*Pparα*$^{-/-}$) mice were obtained from Jackson Laboratory (Bar Harbor, ME). ATF4 liver conditional knockout mice (*Atf4*$^{ΔHep}$) and control (*Atf4*$^{F/F}$) were obtained from Dr. Christopher M. Adams from the University of Iowa. The intestinal epithelial cell (IEC)-specific expression of the constitutively active gp130 variant (*gp130*$^{Act}$) mice was produced previously[58]. All animals were housed at the University of California San Diego (UCSD) Animal Care Facility and received food and water ad libitum.

Due to the tendency for TCS to accumulate in many animals and plants that are consumed by humans, we administer TCS in the chow[21]. Breeding pairs of all mice strains (6 weeks old) were fed with a chow diet containing 0.012% TCS (Sigma-Aldrich, 72779) dissolved in vehicle (2 ml DMSO + 10 ml water) or vehicle alone. The females received TCS in the chow during pregnancy and lactational period (until neonates complete 21 days old). Neonatal mice remained with the breeding pairs for 21 days before being sacrificed, and serum, liver and intestines were collected for analysis. Treatment with TCS did not affect female body weight. For obeticholic acid (OCA, MedChemExpress, HY-12222) experiments, C57/BL6 neonates breastfed TCS were treated with 100 mg/kg of vehicle or OCA (100 mg/kg per day) by oral gavage (p.o.) from day 16 to day 20. The mice were euthanized when they were 21 days old and tissues were collected for further analysis. For all

experiments we used only male neonatal mice from at least two independent litters of each treatment group.

The protocols for mice handling and procedures were approved by the UCSD Animal Care and Use Committee (IACUC), and these protocols were conducted in accordance with federal regulations. Animal Protocol S99100 was approved by the UCSD Institutional Animal Care and Use Committee.

### Triclosan measurements on milk and serum

We analyzed TCS in the breastmilk of 14-day-old neonates. TCS in the serum was analyzed at 21 days. All solvents used for sample preparation and metabolomic analysis were LC/MS grade. Methanol, acetonitrile, water, and isopropanol were purchased from Honeywell International Inc. Ethanol and acetic acid were purchased from Sigma-Aldrich.

All pipetting instruments and consumables were purchased from Eppendorf. LCMS amber autosampler vials and tri-layer vial caps were purchased from Agilent Technologies and 300 µL glass inserts were purchased from Wheaton. Kinetex C18 1.8 mm (100 × 2.1 mm) UPLC columns were purchased from Phenomenex Inc. UPLC BEH RP-18 guard columns were purchased from Waters Inc. Pooled human plasma was obtained from Bioreclamation IVT. Isotopically labeled (13C12) labeled triclosan standard was purchased from Cambridge Isotopes (P/N CLM-6779-1.2).

To each sample, 20 µL of serum and milk was transferred to a clean 1.5 mL microfuge tube. To each sample 80 µL of ethanol extraction solvent containing 250 nM of 13C labeled triclosan was added. Samples were then vortexed at 2000 rpm for 5 min at 4 °C to allow for protein precipitation followed by centrifugation at 14,000 rpm for 5 min at 4 °C. For each sample, 75 µL of supernatant was transferred to an amber glass HPLC vial (P/N 92-5182-0716) containing a Wheaton 300 µL (P/N 11-0000-100) glass insert. Samples were stored at 4 °C in a Thermo Scientific Vanquish UHPLC autosampler until analysis by LC-MS/MS.

LC-MS/MS Data Acquisition was performed as previously described[59,60]. In brief, 20 µL of the sample was injected onto a Phenomenex Kinetex C18 reverse phase column and compounds were eluted with a constant flow rate of 0.375 mL/min using the following gradient: 0-0.25, 99–99% A, 0.25-5 min, 99–45% A, 5–5.5 min, 55–1% A, 5.5–7.5 min,1%A, where mobile phase A is 70:30:0.1 water: acetonitrile: acetic acid and mobile phase B is 50:50:0.02 acetonitrile: isopropanol: acetic acid. Compounds were detected using a Thermo Scientific QExactive Orbitrap mass spectrometer equipped with a heated electrospray ionization (HESI) source operating in negative ion mode with the following source parameters: sheath gas flow of 40 units, aux gas flow of 15 units, sweep gas flow of 2 units, spray voltage of −3.5 kV, capillary temperature of 265 °C, aux gas temp of 350 °C, S-lens RF at 45. Data was collected using an MS1 scan event followed by 4 DDA scan events using an isolation window of 1.0 $m/z$ and a normalized collision energy of 35 arbitrary units. For MS1 scan events, scan range of m/z 225-650, mass resolution of 17.5k, AGC of 1e6 and inject time of 50 ms was used. For tandem MS acquisition, mass resolution of 17.5 k, AGC 5e5 and inject time of 80 ms was used. TCS was identified by matching accurate mass, retention time and MS/MS fragmentation pattern with commercial standard and quantified using the isotopically labeled internal standard. Data was collected using Thermo Xcalibur software (version 4.1.31.9) and analyzed using Thermo QualBrowser (version 4.1.31.9) as well as MZmine 2.36.

### Histology and immunohistochemical procedures

To analyze liver and small intestine (jejunum) morphology, tissue samples were fixed in 10% buffered formalin phosphate (Fisher Chemicals) for analysis at the UCSD Histology Core. Samples were embedded in paraffin, sliced into 5 µm sections and stained with H&E (hematoxylin and eosin). A semi-quantitative summary of the

assessment scores for NAFLD histopathology's were performed[34]. For detection of lipid accumulation, frozen liver samples were embedded in Tissue-Tek optimum cutting temperature (OCT) compound, frozen at dry ice, stored in −80 °C and then sliced to 5 µm sections and stained with Oil Red O (ORO) and counterstained with hematoxylin as previously described[6]. For detection of fibrosis, paraffin-embedded sections were stained with Sirius Red as previously described[22,23]. Representative images were captured on an upright light/fluorescent microscope (Zeiss) equipped with AxioCam camera.

For the staining of F4/80 (ThermoFisher, 41-4801-82) and Ki-67 (GeneTex, GTX16667), paraffin liver sections were prepared in the Histology Core (University of California, San Diego). Formalin-fixed, paraffin-embedded liver slides were deparaffinized and rehydrated, using xylene followed by alcohol and PBS washings. Antigen retrieval of tissue slides and the immunohistochemical staining with a primary antibody, secondary biotinylated antibody, and streptavidin-HRP (Pharmingen) were achieved as described previously[22]. All primary antibodies were diluted 1:100 and secondary antibodies were diluted 1:200. Ki-67-positive cells were counted on five fields of four different mice of each group of ×200 magnification per slide.

Detection of apoptotic cells in tissue sections was performed by the TUNEL assay with the In-Situ Cell Death Detection Kit (TMR red, Roche), according to the protocol described previously[22]. TUNEL-positive cells were counted on five fields of four different mice of each group of ×200 magnification per slide. For Sirius Red and F4/80 quantification, images were converted to RGB stack and middle section was used. Threshold was adjusted to cover the positive stained area and percentage of positive area was calculated by Image J2 Fiji software, v. 2.3.0.

### Reverse Transcription Quantitative PCR

Tissue samples were homogenized in 1 mL TRIzol Reagent (Invitrogen, Waltham, MA) according to the manufacturer's instructions and total RNA was extracted. Using iScript Reverse Transcriptase (Bio-Rad Laboratories, Hercules, CA), 1 µg of total RNA was used for the generation of cDNA in a total volume of 12 µL as outlined by the manufacturer. Following cDNA synthesis, quantitative PCR was carried out on a CFX96 qPCR system (BioRad) by using SsoAdvanced SYBR Green Supermix (BioRad). Primers sequences are provided in Supplementary Dataset.

### RNA sequencing analysis

For RNA-seq studies, each RNA sample consisted of RNA from 3 mice, and 3 RNA samples (a total of 9 mice) per group were analyzed as previously performed in our laboratory. The sequencing library was prepared using the Illumina TruSeq RNA Sample Prep Kit (FC-122-1001; Illumina, San Diego, CA) with 1 ug of total RNA. The sequencing was performed on an NovaSeq 6000 sequencer in the IGM Genomics Center at UCSD. RNA sequencing data analysis was performed by Dr Kristen Jepsen (UCSD). Image deconvolution, quality value calculation, and the mapping of exon reads and exon junctions were performed at the UCSD sequencing core. Base calling was performed using bcl2fastq (v2.17.1.14; Illumina). RNA sequencing reads were aligned Q20 to the mice genome (mm10) with STAR (v2.2.0c)[61] with default parameters, only uniquely alienable reads were used for downstream analysis. Gene expression values were calculated using HOMER by quantifying strand-specific reads across annotated gene exons (RefSeq) and reported as fragments per kilobase of exon per million mapped reads. Sequencing reads were aligned to the Mus musculus (UCSC mm10) genome. Heatmaps were drawn using GraphPad Prism 9.1.0.

### Western blot analysis

For whole tissue analyses, minced liver tissue (0.1 mg) was homogenized in 0.4 mL 1 X RIPA lysis buffer (EMD Millipore, Billerica, MA) supplemented with protease inhibitor cocktail (Sigma-Aldrich). After

homogenization, the samples were centrifuged at 15,000$g$ for 20 min at 4 °C and the supernatants transfer to a new tube and kept at −80 °C until analysis.

Western blots were performed by using NuPAGE 4–12% BisTris-polyacrylamide gels (Invitrogen) with the protocols described by the manufacturer. Protein (30 μg) was electrophoresed at 170 V for 50 min and transferred at 20 V for 2 h to PVDF membranes (EMD Millipore). Membranes were blocked with 5% non-fat milk at room temperature for 1 h and incubated with primary antibodies, at 4 °C overnight. Membranes were washed and exposed to HRP-conjugated secondary antibodies (anti-mice IgG, anti-rabbit IgG or anti-rat IgG) for 1 h at room temperature. Protein was detected by the ECL Plus Western blotting detection system (BioRad) and was visualized by the BioRad Chemidoc Touch Imaging System. Antibodies used for Western Blotting were: GAPDH (Santa Cruz, sc-32233), β-catenin (Santa Cruz, sc-7963), claudin-1 (Santa Cruz, sc-166338), AMPKα1/2 (Santa Cruz, sc-25792), CYP7A1 (Abcam, ab-65596), SCD1 (Santa Cruz, sc-14720), phospho-AMPK (Cell Signaling, CS2535), phospho-eIF2α (Cell Signaling, CS3597), ATF4 (Cell Signaling, CS11815), IL-1β (Cell Signaling, CS12426), cleaved caspase 3 (Cell Signaling, CS9661), FASN (Cell Signaling, CS3180), FOXO1 (Cell Signaling, CS2880), NQO-1 (Cell Signaling, CS62262), cytochrome C (Cell Signaling, CS11940), caspase-9 (Cell Signaling, CS9508S), phospho-β-Catenin (Cell Signaling, CS5651), LGR5 (Abclonal, A10545), SGK1 (Abclonal, A3936). The secondary antibodies anti-mice IgG horseradish peroxidase (HRP) conjugated antibody and anti-rabbit IgG HRP-conjugated antibodies were obtained from Cell Signaling Technology, Inc. (Danvers, MA). All primary antibodies were diluted 1:1,000 and secondary antibodies were diluted 1:3000.

### Primary hepatocytes culture and PERK-targeted small interfering RNA (*siRNA*) Regulation

Primary hepatocytes were isolated from 4-week-old C57/B6 mice. Mice were anesthetized and the portal vein was cannulated and perfused with Hanks' balanced salt solution (without Mg$^{2+}$ or Ca$^{2+}$) followed by perfusion with Hanks' balanced salt solution with Mg$^{2+}$ and Ca$^{2+}$ containing 0.1 mg/mL of Liberase (Roche Applied Science). Following the removal of the liver, the resulting hepatocytes were filtered through a sterile 70-μm filter. The hepatocytes were than cultured in 12-well collagen-treated plates with Dulbecco's modified Eagle's medium (DMEM) medium containing 10% fetal bovine serum (FBS) and penicillin/streptomycin for 6 hours. After changing the medium, the hepatocytes were exposed to TCS at 30 μM with DMEM medium supplemented with 25Mm HEPES, 40 ng/mL Dexamethasone, 1x Insulin-transferrin-selenium and penicillin/streptomycin. After 72 h, hepatocytes were collected for Q-RT-PCR and WB.

siRNA specific for mouse PERK (Santa Cruz, sc-36214) and control (Santa Cruz, sc-37007) were purchased at Santa Cruz Biotechnology. Four hours after primary hepatocytes were isolated from 4- to 6- week-old C57/B6 mice, cells were transfected in the presence of 10 nM of either siRNA or control RNA with Lipofectamine RNAiMAX reagent (Invitrogen) in a final volume of 0.5 mL of OPTI-MEM. After 24 hours medium was changed with fresh medium supplemented with 25Mm HEPES, 40 ng/mL dexamethasone, 1x insulin-transferrin-selenium and penicillin/streptomycin, containing TCS or DMSO. Lipofectamine and siRNA were kept until the end of the experiment. Fourth-eight hours later, cells were used for RNA and protein extraction. Q-RT-PCR were carried out to examine gene expression levels and WB.

### Triglyceride and FGF21 ELISA

FGF21 and triglyceride serum levels were measured using FGF-21 Quantikine enzyme-linked immunosorbent assay kit (R&D System), triglyceride colorimetric assay kit (Cayman Chemicals), respectively. Lipid contents were extracted by using Bligh and Dyer method[62] for analysis in liver tissue using the same ELISA kits previously mentioned.

### Statistical analyses

Data are represented as mean ± SEM. For all data we used Shapiro–Wilk test to verify the normality of data; when data is normally distributed, statistical significance was determined using two-sided Student's $t$ test, otherwise, significance was determined by Wilcoxon–Mann–Whitney test. $P$ values < 0.05 were considered statistically significant, and statistically significant differences are indicated with *$P$ < 0.5; **$P$ < 0.01; ***$P$ < 0.001. Statistical analyses were performed using GraphPad 9.1.0 (San Diego, CA). Exact P values for all comparisons, together with group size for each group, were listed in Supplementary Dataset.

### Reporting summary

Further information on research design is available in the Nature Research Reporting Summary linked to this article.

## Data availability

The data reported in this paper have been deposited in the Gene Expression Omnibus (GEO) database with the accession code GSE200705. All other data generated or analyzed during this study are available from the corresponding author on reasonable request. Source data are provided with this paper.

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

## Acknowledgements

This work was supported by National Institute of Environmental Health Sciences Grants R21-ES031849 (R.H.T.), P42-ES010337 (R.H.T., M.J., and M.K.), NIH grant R21-AI35677 (S.C.), and a Seed grant to R.H.T. made available through the UC San Diego Larsson-Rosenquist Foundation Mother-Milk-Infant Center of Research Excellence (MOMI CORE). The support of the Family Larsson-Rosenquist Foundation is gratefully acknowledged."

## Author contributions

A.A.W., M.J., M.K., and R.H.T. conceptualized the study. A.A.W., X.Y. and E.M. performed animal experiments. A.A.W. and X.Y. performed and analyzed in vitro studies. J.D., J.D.W., and M.J. performed and analyzed LC-MS/MS experiments. A.A.W., M.J., S.J. performed the data analysis. M.J., M.K., S.C., and R.H.T. provide the resources. R.H.T. supervised the study. A.A.W., M.K., and R.H.T. wrote the manuscript.

## Competing interests

The authors declare no competing interests.
