## [Peer Review File · Nature Communications]

Title: Lactational deliver of Triclosan promotes Non-alcoholic Fatty Liver Disease in newborn miceREVIEWER COMMENTS

Reviewer #1 (Remarks to the Author):

Weber et al. describe the effects of Triclosan exposure via lactation on hepatic lipid accumulation in mice. They suggest TCS may underlie recent increases in pediatric NAFLD. Although the results are intriguing, several suggestions are provided that would increase the significance and clarity of the manuscript. First, beyond age of onset, are there any pathobiological or biochemical differences that distinguish adult NAFLD from pediatric NAFLD? Histopathology is still considered the gold standard for diagnosing NAFLD. Therefore, a semi-quantitative summary of the assessment scores for steatosis, necrosis, inflammation, ballooning and fibrosis should be included as well as effects on pup body weights, absolute liver weights and relative liver weights. Histopathology may also indicate the significance of fibrotic, oxidative stress and inflammation gene expression that, although statistically significant, was modest and not compelling in isolation. Likewise, did histological analysis of intestines in TCS exposed mice exhibit any pathobiology suggestive of disrupted structure that would support assertions of damaged intestinal barrier based on caspase increases and decreases in Cldn gene expression. Furthermore, what segment of the intestine was examined. The increase in ALT to ~200 IU/L is modest, and likely will not be reflected in liver damage in histopathology and therefore may warrant qualification within the text. It was also odd that the significance of decreased AMPK-P protein levels was examined by reporting effects on target gene expression as opposed to target protein phosphorylation. Although TCS and PCB126 are reported to repress Fgf21 mRNA levels, it should be noted that dioxin is reported to induce Fgf21 mRNA levels (PMID: 24769090). Quantitative measurements of TCS in the breastmilk and serum of breastfed neonates are appreciated. It would be helpful if concentrations in the Introduction and in the Results were reported in the same units where possible, and if concentrations of TCS in the neonate livers could also be provided. Lastly, it is also important to place these results and concentration levels in perspective with the Dayan paper (reference # 22) since “extensive set of experimental toxicological and pharmacokinetic studies and the analogous clinical investigations in humans have shown that triclosan generally has limited toxic potential.”

Reviewer #2 (Remarks to the Author):

The authors evaluated the short term-effect of TCS exposure, an antimicrobial additive, in pregnant mice under their pups during lactation. TCS concentrations detected in the breastmilk collected from the stomach of the pups as well in the serum were similar to those found in milk and serum from human. The main finding of this manuscript is that exposure to TCS during lactation drives to an early increase in markers for the development of fat liver mediated by the PERK-eIF2 α -ATF4-PPAR α cascade. Data were well-organized in 04 figures. The discussion section is coherent and focused. Literature citations seem adequate. I think this study is interesting, however, there are major issues that need to be addressed before recommending acceptance.

ABSTRACT: The experimental model must be better explained.

INTRODUCTION: The introduction is very focused and with interesting references, but I suggest that the study hypothesis be clearly written at the end of this session.

RESULTS: It would be interesting to present the body weight and liver weight of newborn mice.

METHODS:

- Lines 592-593: Inform the sex of newborn mice at 21 days of age. This is very important, as several studied markers differ between males and females, even before the puberty period.

- Regarding TCS: 1) Justify why the administration was made in the chow; 2) clarify if the maternal diet containing TCS was given only during pregnancy or it was maintained also during lactation. 3) Was there a change in maternal food intake or body weight? 4) Why measure TCS in gastric content (milk) only at 14 days of lactation? Why was TCS not measured before? Why did you not measure the gastric content at 21 days (which reflects what the pups ingested in terms of milk and chow).

- Page 29, lines 583-593: clarify here the "n" used. The correct is to inform how many pups/litter/age (14 or 21) were studied.

FIGURE LEGENDS: inform the "n" according to above comment.

Reviewer #3 (Remarks to the Author):

In this study, the authors investigated the role of lactation triclosan in the development of steatosis in newborn mice and its underlying mechanisms. The findings are interesting and of potential significance to public health, however additional evidences are needed to support the authors' conclusion. Specific points need to be addressed:

1. The authors stated that the effect of TCS is mediated by PERK-eIF2 α -ATF4 signaling. Since p-eIF2 α /ATF4 can be activated by other ISR signaling pathways besides PERK, p-PERK and PERK levels should be examined in all relevant experiments.
2. To investigate if TCS stimulates ER stress, ER stress markers and other unfolded protein response genes, such as Chop, Bip, Xbp1s, Atf6, need to be examined as well.
3. It would be better to examine cell death by TUNEL staining. Is there histological evidence for liver fibrosis and inflammation in TCS-treated mice?
4. The expression levels of p-eIF2 α and ATF4 should be evaluated in PERK inhibitor studies.
5. Additional evidence is needed to conclude that PPAR α acts downstream of ATF4. Does ATF4 directly regulate PPAR α ? Was TCS-induced PPAR α activation abolished in ATF4 null mice or with PERK inhibition?
6. The authors concluded that OCA administration and gp130^{>Act} mice reduce TCS induced

fatty liver and ER stress. Liver steatosis should be evaluated by oil red O in OCA-treated mice and in gp130^{Act} mice. ER stress markers should also be examined in these models.
7. Phospho- protein western blotting should be accompanied by total protein.

Reviewer #4 (Remarks to the Author):

In the present study, the authors determined the effects of breastmilk-derived Triclosan (TCS) in the pathogenesis of pediatric non-alcoholic fatty liver disease (NAFLD). The authors showed that lactational delivered TCS induced early indicators of NAFLD development through a PERK-eIF2 α -ATF4-PPAR α signaling. Furthermore, the activation of intestinal mucosal-regenerative gp130 signaling by OCA rescued TCS-caused liver PPAR α signaling and DNL, which indicated that TCS exerted effects primarily via the intestine. The study is well designed, and the data are convincing and well presented. The authors demonstrated TCS exposure via lactation might play the important role in the onset and progression of pediatric NAFLD, however How TCS takes effects on ER stress or intestinal barrier is still unclear. Particular concerns are noted below:

1. The authors showed the OCA treatment and blocking gp130 signaling improved DNL, should present more detailed data, including the levels of TG in the liver, the HE or Oil red staining of the liver and others.
2. The OCA treatment only influenced the DNL? How about the glucocorticoid response?
3. TCS treatment damaged the intestinal barrier, the expression of Reg3b and Reg3g was upregulated in TCS treatment. However, the research (PMID: 26867181) reported that the reg3b and reg3g protects mice from alcohol-induced steatohepatitis, it seems that the reg3b or reg3g played a beneficial effects.
4. Was the TCS-induced hepatic DNL direct or dependent on an indirect effect mediated by intestinal barrier disruption? The authors suggest to supplement the experiments on the effect of TCS on hepatocytes directly.
5. Did TCS induce hepatocytes injury only without activating immune cell inflammation? How did the Kupffer cells change under TCS treatment?
6. The dose of TCS is need to be investigated more clearly. One statistic cited by the authors is that concentrations of TCS in human breastmilk samples reached up to 2,100 $\mu\text{g}/\text{kg}$ in line 97. In fact, the concentrations of these samples ranged from 0 to 2100 $\mu\text{g}/\text{kg}$ lipid, not $\mu\text{g}/\text{kg}$ milk, which means the average of the concentrations in the 5 samples with the highest levels was calculated 1742 $\mu\text{g}/\text{kg}$ lipid, corresponding to only 35.8 $\mu\text{g}/\text{kg}$ whole breast milk (PMID: 17011099).
7. Under the same treatment conditions, why the fold change of the same genes in WT mice in Fig3a and Fig3g were different?
8. In Fig3g, the author showed that gene Fasn was increased in WT neonatal mice breastfed with TCS milk for 21 days. But in Fig4B of another article (PMID: 33229553), Fasn was decreased in Ppar α wt mice under a HFD-containing TCS administration. Why the results are opposite, with the consistent trend in gene scd1 in both model?
9. Did the TCS concentration in the serum change in the gp130Act mice compared with WT mice under TCS treatment?

10. Where was western-blot results of phosphorylated PERK in figure 1?
11. How did the body weight change in the TCS treatment? Did the TCS increase the mice body weight?
12. The sample sizes of many experiments were insufficient (n=3).

UNIVERSITY of CALIFORNIA, SAN DIEGO

SCHOOL OF MEDICINE

Robert H. Tukey
Professor
rtukey@ucsd.edu

REVIEWER #1 (Remarks to the Author):

Weber et al. describe the effects of Triclosan exposure via lactation on hepatic lipid accumulation in mice. They suggest TCS may underlie recent increases in pediatric NAFLD. Although the results are intriguing, several suggestions are provided that would increase the significance and clarity of the manuscript. First, beyond age of onset, are there any pathobiological or biochemical differences that distinguish adult NAFLD from pediatric NAFLD? Histopathology is still considered the gold standard for diagnosing NAFLD. Therefore, a semi-quantitative summary of the assessment scores for steatosis, necrosis, inflammation, ballooning, and fibrosis should be included as well as effects on pup body weights, absolute liver weights and relative liver weights. Histopathology may also indicate the significance of fibrotic, oxidative stress and inflammation gene expression that, although statistically significant, was modest and not compelling in isolation. Likewise, did histological analysis of intestines in TCS exposed mice exhibit any pathobiology suggestive of disrupted structure that would support assertions of damaged intestinal barrier based on caspase increases and decreases in *Cldn* gene expression. Furthermore, what segment of the intestine was examined. The increase in ALT to ~200 IU/L is modest, and likely will not be reflected in liver damage in histopathology and therefore may warrant qualification within the text. It was also odd that the significance of decreased AMPK-P protein levels was examined by reporting effects on target gene expression as opposed to target protein phosphorylation. Although TCS and PCB126 are reported to repress *Fgf21* mRNA levels, it should be noted that dioxin is reported to induce *Fgf21* mRNA levels (PMID: 24769090). Quantitative measurements of TCS in the breastmilk and serum of breastfed neonates are appreciated. It would be helpful if concentrations in the Introduction and in the Results were reported in the same units where possible, and if concentrations of TCS in the neonate livers could also be provided. Lastly, it is also important to place these results and concentration levels in perspective with the Dayan paper (reference # 22) since "extensive set of experimental toxicological and pharmacokinetic studies and the analogous clinical investigations in humans have shown that triclosan generally has limited toxic potential."

Response: Pediatric NAFLD displays the same morphological lesions observed in adult NAFLD. However, the pattern of distribution of these lesions are different (PMID: 22249728). We have added a paragraph in the introduction to explain this (Lines 77-80). Also included are body and liver weights in supplemental material (Figure S1) and a semi-quantitative histopathology score table (Table S1) and text lines 131-132 and 202-205, respectively. As suggested, we increased the number of samples for phosphorylated AMPK (Figure 4). The segment used to analyze the intestine was jejunum and in histological analysis there was no apparent damage. With regards to ALT levels, the increase was modest, but it is important to remember these mice are only 21 days old. Concerning FGF21, we have demonstrated in adult mice exposed to dietary TCS and a HFD that *Fgf21* is completely repressed (PMID: 33229553). The report by Klassen (PMID: 24769090) does indicate that *Fgf21* can be induced by the AhR. However, TCS does not activate the AhR, indicating selective differential mechanisms influencing *Fgf21* gene regulation. As recommended, we have changed the TCS measurement units in the results for $\mu\text{g}/\text{kg}$. To change the units, we considered the density of milk at 1.04 g/mL and serum 1.02g/mL (Figure 1). Measurements of TCS in the liver were not performed. This experiment would best be performed with radio labelled TCS, but unfortunately, we do not have access to this agent.

REVIEWER #2 (Remarks to the Author):

DEPARTMENT OF PHARMACOLOGY

9500 GILMAN DRIVE, LA JOLLA, CALIFORNIA 92093-0722 Telephone (858) 822-0288 Fax (858) 822-0363

The authors evaluated the short term-effect of TCS exposure, an antimicrobial additive, in pregnant mice under their pups during lactation. TCS concentrations detected in the breastmilk collected from the stomach of the pups as well in the serum were similar to those found in milk and serum from human. The main finding of this manuscript is that exposure to TCS during lactation drives to an early increase in markers for the development of fat liver mediated by the PERK-eIF2 α -ATF4-PPAR α cascade. Data were well-organized in 04 figures. The discussion section is coherent and focused. Literature citations seem adequate. I think this study is interesting, however, there are major issues that need to be addressed before recommending acceptance.

ABSTRACT: The experimental model must be better explained.

Response: We have added in the abstract more detail about the experimental model. Lines 50-53.

INTRODUCTION: The introduction is very focused and with interesting references, but I suggest that the study hypothesis be clearly written at the end of this session.

Response: We have added a statement regarding the working hypothesis at the end of the introduction. Lines 114-122.

RESULTS: It would be interesting to present the body weight and liver weight of the newborn mice.

Response: We added this data on results and supplemental information and lines 131-132 (Supplementary Figure S1).

METHODS:

- Lines 592-593: Inform the sex of newborn mice at 21 days of age. This is very important, as several studied markers differ between males and females, even before the puberty period.

Response: We have added this information in Material and Methods. Lines 705-706.

- Regarding TCS: 1) Justify why the administration was made in the chow; 2) clarify if the maternal diet containing TCS was given only during pregnancy, or it was maintained also during lactation. 3) Was there a change in maternal food intake or body weight? 4) Why measure TCS in gastric content (milk) only at 14 days of lactation? Why was TCS not measured before? Why did you not measure the gastric content at 21 days (which reflects what the pups ingested in terms of milk and chow).

Responses: (1) All of the work that we have been conducting have relied upon TCS distribution in chow. For consistency we have elected to continue with this practice. Also, TCS is more unstable in water than in solid material (PMID: 26738475), lines 694-695.

(2) The maternal diet was given during pregnancy and lactation (Lines 697-699).

(3) TCS increases body weight only in long term exposure (PMID: 25404284). In this study, female body weight didn't change. We added female body weight in Material and Methods (Lines 700-701).

(4) We analyzed milk content at 14 days age to confirm that TCS was being transferred by lactation prior to their ability to eat chow (Lines 713-714). TCS has a cumulative impact. It therefore seemed logical to us to analyze TCS in milk at day 14 and in the serum at day 21 to identify the amount of TCS systemically at the end of the lactation period.

- Page 29, lines 583-593: clarify here the "n" used. The correct is to inform how many pups/litter/age (14 or 21) were studied.

Response: We used at least two litters per experiment (Lines 705-706). The number of mice used are indicated in each figure caption.

FIGURE LEGENDS: inform the "n" according to above comment.

Response: We added the number of neonates in each of the figure legends.

REVIEWER #3 (Remarks to the Author):

In this study, the authors investigated the role of lactation triclosan in the development of steatosis in newborn mice and its underlying mechanisms. The findings are interesting and of potential significance to public health, however additional evidence are needed to support the authors' conclusion. Specific points need to be addressed:

The authors stated that the effect of TCS is mediated by PERK-eIF2 α -ATF4 signaling. Since p-eIF2 α /ATF4 can be activated by other ISR signaling pathways besides PERK, p-PERK and PERK levels should be examined in all relevant experiments.

Response: We didn't analyze WB of PERK and phosphor-PERK in this study. However, we performed a new experiment in primary hepatocytes that highlighted the actions of TCS. Neonatal hepatocytes were transfected with siRNA with PERK in hepatocytes and showed that ATF4 activation is dependent of PERK. The results are in Figure 2 and lines 158-171.

To investigate if TCS stimulates ER stress, ER stress markers and other unfolded protein response genes, such as Chop, Bip, Xbp1s, Atf6, need to be examined as well.

Response: Thank you for this suggestion. We investigated ER stress induced genes, including *Grp78*, *Edem*, *Xbp1s* and *Chop*. All these marker genes were upregulated in TCS treated mice. We have included these studies in Figure 1 and lines 133-138.

It would be better to examine cell death by TUNEL staining. Is there histological evidence for liver fibrosis and inflammation in TCS-treated mice?

Response: Thank you for this suggestion and yes we did carryout TUNEL staining. That data was added to the Supplemental Information (lines 153-155, Figure S1). We also performed Sirius red staining for fibrosis and F4/80 for inflammatory staining. TCS treated neonates showed increase staining (Figure 3, lines 201-202). In support of this, RNA-seq analysis showed an increase in genes related to inflammation and fibrosis.

The expression levels of p-eIF2 α and ATF4 should be evaluated in PERK inhibitor studies.

Response. Thank you for this suggestion. The PERK inhibitor studies have been replaced with studies in primary hepatocytes examining the impact of PERK inhibition by siRNA (Figure 2 and lines 158-171).

Additional evidence is needed to conclude that PPAR α acts downstream of ATF4. Does ATF4 directly regulate PPAR α ? Was TCS-induced PPAR α activation abolished in ATF4 null mice or with PERK inhibition?

Response: ATF4 interacts with many transcriptional factors. In the present study, PPAR α target genes are decreased in *Atf4* ^{Δ Hep} mice when compared to *Atf4*^{F/F} (Lines 237-240 and Supplemental Figure S5). Hepatocytes transfected with PERK specific siRNA leads to blockage of ATF4 and PPAR α target genes (Lines 241-245 and Figure 5). Treatment with TCS in *Atf4* ^{Δ Hep} and *Ppara*^{-/-} shows no lipid accumulation (Figures 5 and 6). Together, these findings results strongly indicate that PPAR α is downstream of ATF4.

The authors concluded that OCA administration and gp130^{Act} mice reduce TCS induced fatty liver and ER stress. Liver steatosis should be evaluated by oil red O in OCA-treated mice and in gp130Act mice. ER stress markers should also be examined in these models.

Response: Thank you for this suggestion. We added these experiments in gp130 mice and OCA treated mice (Figures 7 and 8. Lines 280-284 and 319-327).

Phospho- protein western blotting should be accompanied by total protein.

Response: We changed the figures for total protein for each phospho-protein.

REVIEWER #4 (Remarks to the Author):

In the present study, the authors determined the effects of breastmilk-derived Triclosan (TCS) in the pathogenesis of pediatric non-alcoholic fatty liver disease (NAFLD). The authors showed that lactational delivered TCS induced early indicators of NAFLD development through a PERK-eIF2 α -ATF4-PPAR α signaling. Furthermore, the activation of intestinal mucosal-regenerative gp130 signaling by OCA rescued TCS-caused liver PPAR α signaling and DNL, which indicated that TCS exerted effects primarily via the intestine. The study is well designed, and the data are convincing and well presented. The authors demonstrated TCS exposure via lactation might play the important role in the onset and progression of pediatric NAFLD, however How TCS takes effects on ER stress or intestinal barrier is still unclear. Particular concerns are noted below:

The authors showed the OCA treatment and blocking gp130 signaling improved DNL, should present more detailed data, including the levels of TG in the liver, the HE or Oil red staining of the liver and others.

Response: We added these experiments in gp130 mice and the OCA experiments (Figures 7 and 8. Lines 280-284 and 319-327).

The OCA treatment only influenced the DNL? How about the glucocorticoid response?

Response: In the absence of TCS treatment, OCA increases expression of glucocorticoid receptor target genes. However, in the presence of TCS these genes are not upregulated when mice are treated along with OCA (Figure S7 and lines 285-293).

TCS treatment damaged the intestinal barrier, the expression of Reg3b and Reg3g was upregulated in TCS treatment. However, the research (PMID: 26867181) reported that the reg3b and reg3g protects mice from alcohol-induced steatohepatitis, it seems that the reg3b or reg3g played a beneficial effect.

Response: Reg3b and Reg3g are induced when mice are challenged and suffering barrier disruption. In this case, Reg3 genes are induced as a compensatory effect of TCS injury.

Was the TCS-induced hepatic DNL direct or dependent on an indirect effect mediated by intestinal barrier disruption? The authors suggest to supplement the experiments on the effect of TCS on hepatocytes directly.

Response: We treated hepatocytes with 30 μ m TCS for 72 hours and DNL, ER stress and PPAR α targets genes are upregulated. We included in the manuscript as Figure 2 and lines 158-171.

Did TCS induce hepatocytes injury only without activating immune cell inflammation? How did the Kupffer cells change under TCS treatment?

Response: There is an activation of immune cells. We did F4/80 immunohistochemistry and TCS treatment increases the number of Kupffer cells in the liver (Figure 3 and lines 201-202).

The dose of TCS is need to be investigated more clearly. One statistic cited by the authors is that concentrations of TCS in human breastmilk samples reached up to 2,100 μ g/kg in line 97. In fact, the concentrations of these samples ranged from 0 to 2100 μ g/kg lipid, not μ g/kg milk, which means the average of the concentrations in the 5 samples with the highest levels was calculated 1742 μ g/kg lipid, corresponding to only 35.8 μ g/kg whole breast milk (PMID: 17011099).

Response: We have altered the units for μ g/kg to be in the same units as referenced in the introduction (Figure 1). To do this we used the density of milk as 1.04 and serum 1.02. We changed is in the introduction with the concentration in the lipids, not in milk.

Under the same treatment conditions, why the fold change of the same genes in WT mice in Fig3a and Fig3g were different?

Response: The WT mice used in *Atf4* and *PPARα* experiments are not the same mice. For the ATF4 experiments we used *Atf4^{F/F}* and *Atf4^{ΔHep}* +/- TCS. In *PPARα* experiments we used as WT C57/B6 mice. That is the reason the fold change is not the same for both experiments. We split the figures. Now the *Atf4* experiment is Figure 5 and *PPARα* experiment is in Figure 6.

In Fig3g, the author showed that gene *Fasn* was increased in WT neonatal mice breastfed with TCS milk for 21 days. But in Fig4B of another article (PMID: 33229553), *Fasn* was decreased in *Ppara* wt mice under an HFD-containing TCS administration. Why the results are opposite, with the consistent trend in gene *scd1* in both model?

Response: Our laboratory in past studies with TCS used a higher concentration compared to the concentration used in the present study. Thus, in past studies, when we used more concentrated TCS, there was a blockage in the DNL pathway. However, in the same way as in the current study, it was possible to observe accumulation of lipids in the liver. We still don't understand how this works but this same result is observed for other toxicants such as bisphenol A (BPA). Less concentrated doses of BPA increase DNL while more concentrated doses block this process (PMID: 21932408). It should also be noted that the impact of lactational TCS exposure in neonates may behave differently in neonates as a result of developmental changes.

Did the TCS concentration in the serum change in the *gp130^{Act}* mice compared with WT mice under TCS treatment?

Response: Unfortunately, we don't know the answer to this. However, we did several other experiments in *gp130^{Act}* mice. First, the levels of TG in this model following TCS exposure are much lower than in WT mice. Second, there is no accumulation of lipid in liver sections from TCS treated *gp130^{Act}* mice (Lines 319-320 and Figure 8)

Where was western-blot results of phosphorylated PERK in figure 1?

Response: We have not analyzed WB of PERK and phospho-PERK in the study. However, we have performed an siRNA inhibition study of PERK in primary hepatocytes demonstrating that ATF4 activation is dependent upon PERK (Figure 2 and lines 158-171).

How did the body weight change in the TCS treatment? Did the TCS increase the mice body weight?

Response: The body weight and liver weight did not change under TCS treatment. We put this data in Supplemental Data, Figure S1 and lines 131-132.

The sample sizes of many experiments were insufficient (n=3).

Response: We increased sample size from all experiments possible. We have indicated this in each experiment in the figure captions.

REVIEWER COMMENTS

Reviewer #1 (Remarks to the Author):

Taken together, in vivo gene expression data, histopathology and ALT in Fig. 1 are not sufficiently compelling to support conclusion of ER stress or DNL following TCS exposure. Likewise, differential gene expression in Fig. 3b and 4b is modest. In contrast the induction of Cd36 (Fig 3) suggests modest fat accumulation may be due to increased dietary uptake. Again, a decrease in P-AMPK would decrease the level of phosphorylation of protein targets, not repress target gene expression. The significance of the repression of gene expression in relation to TSC-induced phenotypes is not discussed. TSC-induced effects in primary hepatocytes were more convincing but the relevance of the TSC concentration is not addressed. Further information is required to interpret Fig 3d. Figures should identify portal triad and central vein, areas of fibrosis and inflammation foci. F4/80 image does not suggest inflammation. A description of how positive areas were calculated is required. The authors also failed to qualify there conclusions in the context of other studies.

Reviewer #2 (Remarks to the Author):

I have nothing more to ask.

Reviewer #3 (Remarks to the Author):

The authors have addressed most of the concerns of the reviewer. The manuscript has significantly improved. It has come to the reviewer's attention that a recent paper showed that the ATF4 antibody from Cell Signaling (#11815) detects nonspecific band in mouse liver extracts. (Reference: Hepatic mTORC1 signaling activates ATF4 as part of its metabolic response to feeding and insulin. PMID: 34303878. Supplementary Figure S6). This is the antibody used in this manuscript. The authors should re-examine this antibody using proper positive and negative controls, and perhaps validate their findings using the BioLegend antibody. In addition, are there better images for Figure 3d and Supplementary Figure S1d? The current images are hard to interpret.

Reviewer #4 (Remarks to the Author):

I had no further comments.

REVIEWER #1 (Remarks to the Author):

Taken together, in vivo gene expression data, histopathology and ALT in Fig. 1 are not sufficiently compelling to support conclusion of ER stress or DNL following TCS exposure. Likewise, differential gene expression in Fig. 3b and 4b is modest. In contrast the induction of Cd36 (Fig 3) suggests modest fat accumulation may be due to increased dietary uptake. Again, a decrease in P-AMPK would decrease the level of phosphorylation of protein targets, not repress target gene expression. The significance of the repression of gene expression in relation to TSC-induced phenotypes is not discussed. TSC-induced effects in primary hepatocytes were more convincing but the relevance of the TSC concentration is not addressed. Further information is required to interpret Fig 3d. Figures should identify portal triad and central vein, areas of fibrosis and inflammation foci. F4/80 image does not suggest inflammation. A description of how positive areas were calculated is required. The authors also failed to qualify their conclusions in the context of other studies.

To address the issues pertaining to R#1 comments, we have broken down this paragraph and have addressed each concern separately.

- Taken together, in vivo gene expression data, histopathology and ALT in Fig. 1 are not sufficiently compelling to support conclusion of ER stress or DNL following TCS exposure.

Response: The exposure to TCS promotes ER stress and DNL. This has been confirmed both in vivo and in vitro. The progression to NASH depends on multiple events. Among these events, hepatic DNL and ER stress are important for disease progression (PMID: 32839596; PMID: 25132496; PMID: 30220454). ER stress response is characterized by the activation of three major ER sensors: IRE1 α , PERK, ATF6, after BIP/GRP78 dissociation and direct association with unfolded/misfolded protein. Our data shows upregulation in mRNA levels of Grp78, the first protein to respond in the ER stress-induced cascade. In addition, ER-stress target genes *Edem1*, *XPB1s* (not statistically) and *Chop* are also upregulated in TCS mice. In corroboration with qPCR findings, we also show increased p-eIF2 α and ATF4 protein levels in TCS mice liver lysates compared to vehicle group. Together, our data showed upregulation of target genes and proteins of ER stress in the TCS group, compared to vehicle (control group). Osowski & Urano, 2011, in their book chapter entitled *Measuring ER Stress and the Unfolded Protein Response Using Mammalian Tissue Culture System* (PMID: 21266244), describes several techniques to measure/analyze ER stress in mammalian cells and tissues. In section 5 they describe the most used methods to study ER stress, including qPCR and Western Blot for the same targets we have used in our work. Elevated liver enzymes indicate inflammation or damage to cells in the liver. Inflamed or injured liver cells leak higher than normal amounts of liver enzymes into the bloodstream. In TCS mice, we showed a 2-3 fold elevation in serum levels of ALT compared to the control group, indicating that TCS intake is causing liver injury. Increases in ALT is a great indicator of liver damage during NAFLD and it's been used extensively as a liver damage biomarker (PMID: 31899206). In addition, serum and liver levels of triglycerides and serum palmitate are also important biomarkers in NAFLD patients (PMID: 32978374; PMID: 32839596). All these events are occurring in neonates after exposure to TCS through lactation.

- Likewise, differential gene expression in Fig. 3b and 4b is modest.

Response: Even if the response is modest, it is still important. These responses are in the early stages of developing NAFLD, so large changes would not be expected. What is clearly important is that the response between the genes are statistically significant, validating our hypothesis that the early stages of inflammation, fibrosis, and oxidative stress are being activated. It is clear both at the protein and transcriptional levels that AMPK and its target genes are being impacted by TCS exposure.

- In contrast the induction of Cd36 (Fig 3) suggests modest fat accumulation may be due to increased dietary uptake.

Response: CD36 and other fatty acid uptake genes are upregulated by TCS treatment. The increase in serum free fatty acids, such as palmitate, can increase hepatic export genes and promote an influx to the liver (PMID: 32978374). Export free fatty acid from the periphery (adipose tissue) and DNL are hallmarks of NAFLD (PMID: 32978374). Both processes are upregulated by TCS treatment. Regarding dietary uptake, BW and LW data showed no difference between groups, so we don't believe that dietary uptake is influenced by TCS treatment.

- Again, a decrease in P-AMPK would decrease the level of phosphorylation of protein targets, not repress target gene expression.

Response: Regarding p-AMPK, we have identified that the GR pathway was downregulated and AMPK can be regulated by GR via *Fgf21*. As we have shown in our study *Fgf21* is downregulated by TCS treatment. This result indicates that the downregulation of *Fgf21* can decrease AMPK target genes. We added a paragraph in the discussion about the downregulation of AMPK target genes (Lines 368-376).

- The significance of the repression of gene expression in relation to TSC-induced phenotypes is not discussed.

Response: We added a paragraph in the discussion about the repression of AMPK target genes in NAFLD. Lines 366-373.

- TSC-induced effects in primary hepatocytes were more convincing but the relevance of the TSC concentration is not addressed.

Response: We have conducted dose-dependent experiments in primary hepatocytes with a host of different TCS concentrations, both below and above the 30 μ M range that we show in this figure. Higher concentrations start to lead to cell death, so we elected to show the results with 30 μ M.

- Further information is required to interpret Fig 3d. Figures should identify portal triad and central vein, areas of fibrosis and inflammation foci. F4/80 image does not suggest inflammation. A description of how positive areas were calculated is required.

Response: We have changed Figure 3d and added new figures showing with central vein and the possible fibrosis areas with F4/80-stained cells. In addition, we have added in supplemental data other F4/80 and Sirius red pictures to visualize the patterns of fibrosis and immune infiltration (Supplemental Figure S4). We have also added the quantification methodology of Sirius red and F4/80 in lines 790-794. After 21 days of exposure, we observed early signs of inflammation including increased F4/80-stained cells, IL1 β protein levels and other inflammatory genes. We have altered the text and changed the term "inflammation" to "immune infiltration" for F4/80 staining, since this is a more appropriate description of this process (Lines 201-203).

- The authors also failed to qualify their conclusions in the context of other studies.

Response. Politely, we disagree with this assessment. This manuscript details the possibility that environmental toxicant exposure through lactation can start to promote NAFLD shortly after birth. In combination with continued exposure coupled with dietary challenges, we argue that Pediatric NAFLD may result in part as a result of early toxicant exposure. In addition, there are very limited examples linking toxicant exposure to models of Pediatric NAFLD.

REVIEWER #2 (Remarks to the Author):

DEPARTMENT OF PHARMACOLOGY

9500 GILMAN DRIVE, LA JOLLA, CALIFORNIA 92093-0722 Telephone (858) 822-0288 Fax (858) 822-0363

I have nothing more to ask.

REVIEWER #3 (Remarks to the Author):

The authors have addressed most of the concerns of the reviewer. The manuscript has significantly improved. It has come to the reviewer's attention that a recent paper showed that the ATF4 antibody from Cell Signaling (#11815) detects nonspecific band in mouse liver extracts. (Reference: Hepatic mTORC1 signaling activates ATF4 as part of its metabolic response to feeding and insulin. PMID: 34303878. Supplementary Figure S6). This is the antibody used in this manuscript. The authors should re-examine this antibody using proper positive and negative controls, and perhaps validate their findings using the BioLegend antibody. In addition, are there better images for Figure 3d and Supplementary Figure S1d? The current images are hard to interpret.

- Concerns about the ATF4 antibody.

Response: The paper cited by the reviewer used the same antibody, but according to the paper in Figure S6, there is no unspecific banding pattern for this antibody—just one band around 50 kDa. However, in this reference, *Atf4* null cells showed some ATF4 protein expression. Figure S5, from the present study, showed that the antibody is specific because there is no protein expression in *Atf4*^{Hep} livers. This image already validates this antibody. In addition, we have validated this antibody in other studies (Proc Natl Acad Sci U S A. 2020 Dec 8;117(49):31259-31266. doi: 10.1073/pnas.2017129117. Epub 2020 Nov 23. PMID: 33229553). All the WB images are in the data source. Additional evidence of the antibody's effectiveness is that it has already been used in more than 500 articles, 200 of them were used for WB, according to the Cell Signaling website.

- Regarding the Figure 3d

Response: We have changed the images for better ones. For Figure S1d, unfortunately we don't have better pictures. But it is possible to see the Ki-67-stained cells. We have explained in greater detail better how the calculations of Figure 3d were performed (Lines 790-794).

REVIEWER #4 (Remarks to the Author):

I had no further comments.

REVIEWERS' COMMENTS

Reviewer #1 (Remarks to the Author):

Following further review of the references provided by the authors, modest gene expression changes <2-fold have previously been reported to suggest changes in ER stress and DNL. In addition, these results are consistent with the debatable increases in steatosis, inflammation and fibrosis. These results are also consistent with the more convincing in vitro results.

Further information about the scale used in Table S1 is requested – specifically what was the range for each lesion (0-5?). This should be accompanied by a written description of scale (for example 1 = less than 5% hepatocytes exhibiting macro/microvacuolization; 5 = greater than 50% hepatocytes exhibiting macro/microvacuolization) as a footnote to Table S1. Lines 197-205 should include there was no evidence of hepatocyte ballooning or necrosis, and that the score was 3/15 (dependin on the scale used).

Reviewer #3 (Remarks to the Author):

I have no further comments.

Point-by-point response to the reviewers' comments

REVIEWER #1 (Remarks to the Author):

Further information about the scale used in Table S1 is requested – specifically what was the range for each lesion (0-5?). This should be accompanied by a written description of scale (for example 1 = less than 5% hepatocytes exhibiting macro/microvacuolization; 5 = greater than 50% hepatocytes exhibiting macro/microvacuolization) as a footnote to Table S1. Lines 197-205 should include there was no evidence of hepatocyte ballooning or necrosis, and that the score was 3/15 (dependin on the scale used).

Response: We added in table S1 the description in the footnote of each lesion according to methodology described by Blunt, 2005. In addition, we added in the main manuscript information about lesions (Lines 198-202).